# Endogenous opioids regulate social threat learning in humans

Jan Haaker[1,2], Jonathan Yi[1], Predrag Petrovic[1] & Andreas Olsson[1]

Many fearful expectations are shaped by observation of aversive outcomes to others. Yet, the neurochemistry regulating social learning is unknown. Previous research has shown that during direct (Pavlovian) threat learning, information about personally experienced outcomes is regulated by the release of endogenous opioids, and activity within the amygdala and periaqueductal gray (PAG). Here we report that blockade of this opioidergic circuit enhances social threat learning through observation in humans involving activity within the amygdala, midline thalamus and the PAG. In particular, anticipatory responses to learned threat cues (CS) were associated with temporal dynamics in the PAG, coding the observed aversive outcomes to other (observational US). In addition, pharmacological challenge of the opioid receptor function is classified by distinct brain activity patterns during the expression of conditioned threats. Our results reveal an opioidergic circuit that codes the observed aversive outcomes to others into threat responses and long-term memory in the observer.

---

[1] Department of Clinical Neuroscience, Karolinska Institute, Stockholm 171 76, Sweden. [2] Department of Systems Neuroscience, University Medical Center Hamburg-Eppendorf, Martinistreet 52, 20246 Hamburg, Germany. Correspondence and requests for materials should be addressed to J.H. (email: j.haaker@uke.de).

In humans and many other species, survival depends on information learned from others about what is potentially dangerous in the environment. Across the life-span, expressions of pain and fear in individuals in our proximity are used as visual cues to associatively learn and predict what should be avoided[1,2]. Yet, the underlying neurochemistry that regulates how others' aversive experiences are translated into our own responses of fear and defense remains unexplored.

Previous research has revealed that defensive responses elicited by directly experienced aversive events are regulated by the release of endogenous opioids[3–5]. Moreover, endogenous opioids play a central role in predicting future occurrence of aversive events in Pavlovian fear conditioning in both rodents and humans[6–8]. During Pavlovian fear conditioning, an association between a conditioned stimulus (CS) and a directly experienced aversive stimulus (US) is learned through repeated exposure to the pairing of these two stimuli. The substrate of such association underlies a neural signal of the US that induces potentiation of CS responsive neurons to the lateral part of the amygdala[9,10]. Release of endogenous opioids in the amygdala and the periaqueductal gray (PAG) reduces the neural teaching signal derived from the US[8,11], which is pivotal for the establishment of a CS–US association[9,10]. In support of this, the blockade of opioid receptors enhances processing of painful USs, strengthens the learning of the CS–US association, and leads to a stronger (drug-free) expression of defensive responses towards the CS[8,12]. Yet, it is unclear if this opioidergic circuitry is also involved in learning from observing another individual's expression of pain, in other words; through an observational US.

It has been argued that a teaching signal derived from observing the outcomes of others' actions dynamically update the observer's own expectations and predictions via associative learning processes[13–17]. In line with this assumption, it has been shown in a range of species that observing conspecifics responses to CS–US pairings leads to defensive responses towards the CS in the observer[18–22], involving activity in the amygdala[18,23]. However, the neural circuitry that processes the observational US during learning is not well understood, and its underlying neuropharmacological basis has not been explored. One possibility is that direct and observational threat learning involve similar neurotransmitter system to compute a teaching signal that orchestrates defensive responses to the CS. This conjecture suggests a role of the endogenous opioid system in observational threat learning. Here, we used pharmacological blockade of opioid receptors (administration of 50 mg Naltrexone or Placebo, randomized, double-blind) to investigate the influence of endogenous opioids on neural signals during observational fear conditioning. We hypothesized that opioid receptor function shapes the teaching signal derived through the observational US. More specifically, we conjectured that opioid receptor blockade would enhance neural signalling (measured using functional magnetic resonance imaging (fMRI)) of the observational US in the PAG and amygdala, leading to stronger long-term expression of conditioned responses.

Forty-three participants underwent an observational fear conditioning procedure (for an overview see Fig. 1a–c), adapted from previous studies[21,23,24], involving an Observational learning stage of threats followed by an Immediate expression test of conditioned threat responses. Seventy-two hours after learning, long-term conditioned memory was tested in the drug-free participants (long-term expression test). Participants were attached to SCR and shock electrodes during all stages. During the Observational threat learning stage, participants watched a video depicting a demonstrator presented with a series of blue and yellow squares serving as CSs. One of the CSs (CS +) was pseudorandomly paired with a shock to the wrist of the demonstrator, who responded by twisting the arm and expressing a brief wince (Observational US) in 12 of the 24 CS + trials (Fig. 1b). The other CS (CS −) was never paired with an observational US. During both the immediate, and long-term, expression test, CSs were presented directly to the participants in absence of the demonstrator and never followed by a US (Fig. 1c). Therefore, conditioned responses might have been extinguished during the test stages. While the immediate test stage allows us to test the expression of learning immediately after acquisition, the long-term test stage examines the persistence/return of the acquired threat associations that are learned via observation. Hence, the participants never experienced the US themselves, enabling us to draw conclusions about learning social in nature.

We found that blockade of opioid receptors during observational fear conditioning enhanced signalling in the amygdala, PAG and midline thalamus and led to stronger long-term expression of conditioned threat responses. Our results provide initial evidence for an opioidergic circuit that transmits aversive learning through observation of others.

## Results

**Opioid receptor blockade enhances threat responses.** In agreement with previous studies, analysis of skin conductance responses (SCRs, see Methods for exclusion criteria) during the Immediate expression test indicated successful observational fear conditioning, that is, CS discrimination, reflected as enhanced SCRs to the CS + as compared to the CS − (repeated measurement analysis of variance (ANOVA) (Naltrexone $N = 18$, Placebo $N = 15$, main effect of stimulus: $F(1,31) = 5.215$; $P = 0.029$; eta$^2 = 0.144$; pair-wise comparison, CS + > CS − : $P = 0.029$; see Supplementary Fig. 1a). We found no differences between groups (main effect of group or factor interaction with group: $P > 0.6$; CS discrimination, that is, CS + > CS −, between groups did not differ: one-sided unpaired t-test, $t(31) < 1$; $P = 0.319$; see Supplementary Table 1) during the immediate expression test, consistent with subtle effects of Naloxone on the expression of conditioned fear during direct conditioning[6]. Then, we tested our hypothesis if blockade of opioid receptors during observational fear learning enhanced long-term expression of threat responses (that is, enhanced CS discrimination). Indeed, the Naltrexone group expressed greater threat responses as compared to the Placebo controls in the drug-free long-term expression test [Repeated measurement ANOVA (Naltrexone $N = 22$, Placebo $N = 20$, stimulus by group interaction: $F(1,40) = 3.713$; $P = 0.061$; eta$^2 = 0.085$; t-test one-tailed of CS discrimination between groups, $t(40) = 1.927$; $P = 0.030$, see Fig. 2a and Supplementary Table 2, as well as Supplementary Fig. 1b for CS-specific responses].

Next, we tested whether brain responses during observational fear learning were related to the psychophysiological expression of threat responses (SCRs, CS + > CS −) during the Long-term test. Using multiple regression, we found that the individual threat responses in the Naltrexone group during Long-term expression test were predicted by activity within the amygdala towards the social US during observational learning of threats (Naltrexone: Pearson's correlation coefficient ($N = 22$), $r = 0.650$, $P = 0.001$, Fig. 2b). This was not the case in the Placebo group (Pearson's correlation coefficient ($N = 20$) $r = -0.059$, $P = 0.8$) and this difference between groups was significant (asymptotic z-test of Fisher transformed coefficients (Naltrexone $N = 22$, Placebo $N = 20$)[25], $Z = 2.538$, $P = 0.011$). We then tested whether the amygdala was responsive to the observational US across both groups and whether these responses were enhanced in the Naltrexone group. For that purpose, we contrasted responses across both groups towards the observational US (occurring at

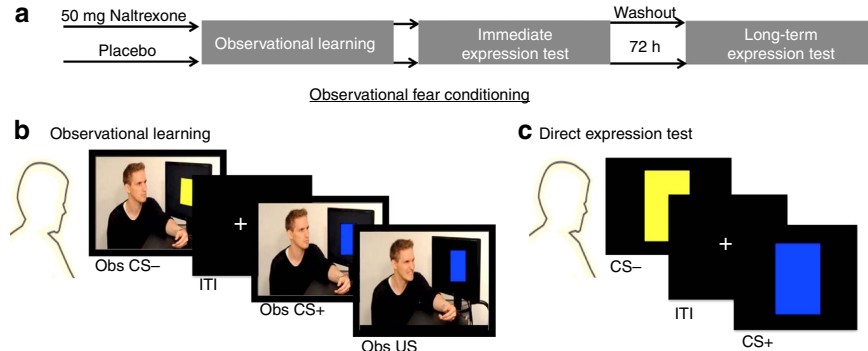

**Figure 1 | Design overview.** (**a**) Participants underwent an observational fear conditioning procedure consisting of an observational learning of threat and two expression test stages examining the conditioned threat responses; immediate expression test and Long-term expression test.(**b**) During the Observational learning stage, participants watched a video depicting a demonstrator presented with conditioned stimuli, CS (coloured squares). In half of the CS + trials, the demonstrator received a shock to the wrist and briefly expressed pain (the observational US for the participant). The other CS (CS − ) was never paired with an observational US. (**c**) Both expression tests for conditioned responses (immediate and long-term) employed direct presentations of the CS + and the CS − to the participant, in the absence of the demonstrator and without any aversive stimulation.

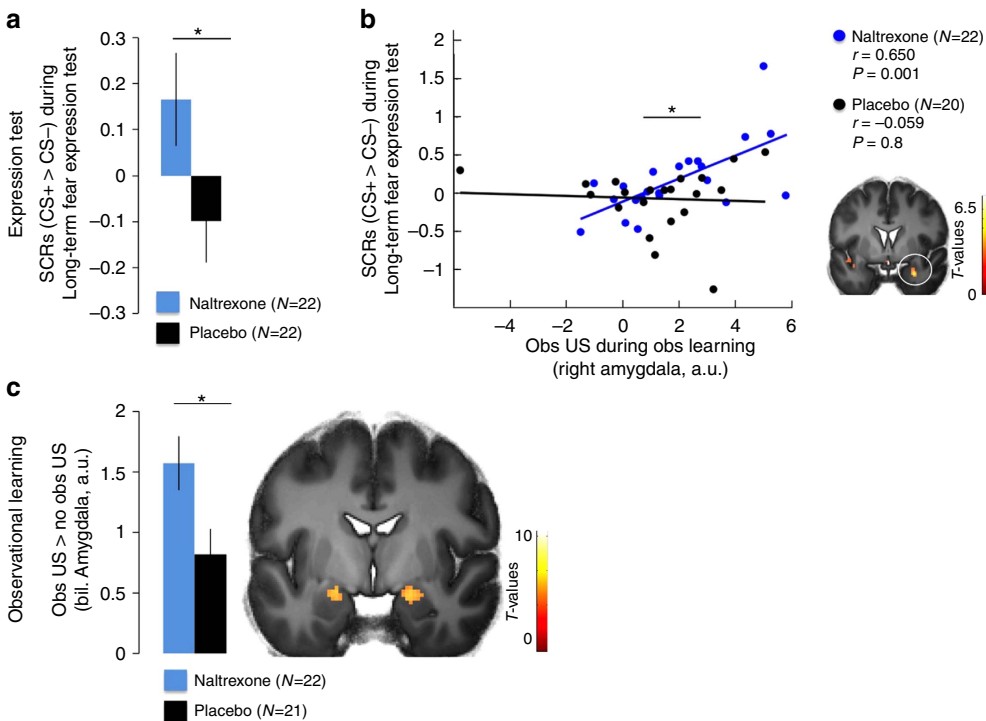

**Figure 2 | Long-term expression test and observational learning.** (**a**) Individuals receiving Naltrexone during observational threat learning showed enhanced conditioned fear responses (SCR, CS + >CS − ; see Supplementary Fig. 1 for CS-specific responses) in the drug-free Long-term test. (**b**) Amygdala responses to the observational US predicted enhanced psychophysiological fear responses in the Naltrexone, but not the Placebo, group 72 h later (full factorial contrast (Naltrexone N = 22, Placebo N = 21), x,y,z (MNI) = 28;2; − 24; t = 3.82; P(SVC) = 0.039). (**c**) Amygdala responses to the observational US were enhanced in the Naltrexone group as compared to Placebo controls, see Supplementary Fig. 3 for specific response to obs US and no obs US trials. 'Obs US' refers to responses to the observational US and 'no obs US' to responses to CS + outcomes that are not followed by the US. The error-bars denote the standard error of the mean and T-maps are superimposed on an average structural image with a threshold of P(FWE, whole brain) < 0.05 for illustrative purposes. Asterisks indicate significant differences between groups.

the end of 50% of the CS + trials, (termed obs US) with responses during the same time point to CS + trials not followed by the US (termed no obs US), which controls for the influence of the preceding CS on observational US responses. This analytic approach, which has previously been used to study US responses during Pavlovian fear conditioning[26], will be employed in all of the following analyses. This contrast revealed that several regions were responsive to the observational US across groups including the bilateral amygdala (full factorial contrast (Naltrexone N = 22,

Placebo N = 21), P(FWE, whole brain) < 0.05, see Supplementary Table 5). The comparison of averaged parameter estimates in the bilateral amygdala region of interest (ROI) between groups revealed a main effect of stimulus (repeated measurement ANOVA (Naltrexone N = 22, Placebo N = 21), F(1,41) = 59.795, P < 0.001; obs US > no obs US), as well as a stimulus by group interaction (F(1,41) = 5.967, P < 0.019), representing a higher differential response to the observational US (obs US > no obs US) in the Naltrexone, as compared to the Placebo group (see

Fig. 2c and Supplementary Fig. 3 for condition-specific responses).

Taken together, these results indicate that the amygdala is responsive towards the observational US during observational fear learning under normal conditions, and that this responsivity can be enhanced by the blockade of opioid receptors. Such an enhancement was moreover associated with persistent (that is, long term) expression of threat responses.

**Opioid receptors scale observational US responses.** The first set of analyses confirmed our hypothesis of higher long-term expression of threat responses in the Naltrexone group. Moreover, increased threat responding was associated with enhanced amygdala responses towards the US during observational learning. Next, we tested whether other regions followed the pattern observed in the amygdala, displaying enhanced coding of the observational US through opioid receptor blockade. We contrasted hemodynamic activity towards the observational US between groups (Naltrexone > Placebo), which revealed higher responses to the observational US in the Naltrexone as compared to the Placebo group in the PAG ROI [Full factorial contrast (Naltrexone $N = 22$, Placebo $N = 21$, left $x,y,z$ (MNI) = $-8$; $-32$; $-8$; t = 3.15; $P(SVC) = 0.016$, see Fig. 3a,b; Supplementary Fig. 7). This activation was located on the left side ventrally to the central aqueduct, most likely located in the PAG (see Supplementary Fig. 7) and extending into the colliculus. In addition, we found a cluster in the midline thalamus (left $x,y,z$ (MNI) = $-6$; $-26$;0; $t = 3.07$; $P(SVC) = 0.038$, see Fig. 3a,b), as well as in the amygdala (as indicated by the previous analysis) and medial prefrontal areas see Supplementary Table 6). Mid-brain-specific normalization and resampling to 1 cubic mm (see Methods) of the estimated responses confirmed the location of these clusters, which have been previously found to be involved in opioid dependent processing of USs during Pavlovian conditioning in rodents and humans[6,8,11]. Interestingly the midline thalamus has been found to be involved in observational fear learning in mice, as well[18]. In addition, studies in both species have highlighted the importance of the PAG for learning to predict aversive events, and have shown that responses towards directly experienced decrease over the time-course of learning[27–29]. On the basis of these findings, we tested whether the (on average) reduced responses in the PAG in the Placebo as compared to Naltrexone group resulted from a difference in temporal dynamics. Indeed, repeated measures ANOVA of observational US responses extracted over trials (averaged blocks over 4 trials) from the PAG revealed that responses decreased in Placebo controls, but not in the Naltrexone group (repeated measurement ANOVA (Naltrexone $N = 22$, Placebo $N = 21$, Block by group interaction: $F(2,82) = 4.88$; $P = 0.011$, quadratic change over blocks $F(1,41) = 7.15$; $P = 0.011$; post hoc two-tailed $t$-test; block 2: $P = 0.002$ and block 3: $P = 0.029$, corrected for multiple comparisons; see Fig. 3c, lower panel). The ANOVA of the extracted responses in the midline thalamus and the left amygdala did not reach significance (repeated measurement ANOVA (Naltrexone $N = 22$, Placebo $N = 21$, Block by group interaction: mid thalamus $F(2,82) = 2.3$; $P = 0.10$; left amygdala $F(2,82) = 2.7$; $P = 0.08$); however, block-wise comparisons between groups, revealed higher responses towards the observational US in block 2 in both regions ($t$-test two-tailed, mid thalamus: $P = 0.002$; left amygdala: $P = 0.004$) and trend-wise in block 3 in the thalamus only ($P = 0.084$, all post hoc tests corrected for multiple comparisons; see Fig. 3c upper panel and Supplementary Fig. 4) mirroring the temporal dynamics of the PAG responses.

These results suggest that PAG responses towards an observational US decrease over trials via opioid receptor activity (Placebo group). As mentioned earlier, this result is in line with previous studies during direct experience of aversive USs in rodents and humans, where a PAG response that decrease over trials was considered as 'teaching signal' enabling associative learning[7,27–32] (see Supplementary Note 2 for additional temporal modelling of PAG responses).

The decrease of US related signalling was also found to limit learned CS responses, whereas the blockade of opioid receptors sustained US signalling (that is, enhancing the teaching signal), which resulted in enhanced responses to the learned CS[8,33]. Hence, we tested if the sustained responses towards the observational US in the PAG in the Naltrexone group enhanced learned anticipatory responses (SCRs) towards the observed CS +. Logistic linear mixed model regression (Supplementary Methods) revealed that PAG responses predicted the SCRs towards the observational CS + (Logistic mixed model regression, Type III Sum of Squares ANOVA (Naltrexone $N = 18$, Placebo $N = 15$), $F(1,94): 4.10$; $P = 0.045$, no significant effect for the SCRs towards the CS −, see Supplementary Tables 3 and 4; Supplementary Fig. 5). Moreover, this positive relationship was trend-wise stronger in the Naltrexone, as compared to, the Placebo group (group by stimulus interaction: $F(1,94): 3.66$; $P = 0.059$). In addition, as compared to the Placebo group, the Naltrexone group showed higher responses in the amygdala towards the observational CS + contrasted with the observational CS − (full factorial contrast (Naltrexone $N = 22$, Placebo $N = 21$), $P(FWE) = 0.043$, see Supplementary Table 7).

Our results suggest that the temporal dynamics of PAG responses elicited by the observation of aversive events are sensitive to opioid receptor function, that is, blockade of opioid receptors sustains PAG responses towards aversive outcomes for others. Importantly, the magnitude of these vicarious responses in the PAG predicted larger anticipatory responses to the learned predictor, that is, PAG responses to the observational US coded the predictive learning of social threat cues.

To test whether the temporal dynamic of PAG responses towards the observational US was functionally connected with other brain regions in the Naltrexone group, we compared condition-specific connectivity (psychophysiological interaction, PPI; Supplementary Methods) of PAG responses towards the observational US between groups. This analysis revealed enhanced coupling of PAG responses with activity in the superior temporal sulcus (STS; Fig. 3d) in the Naltrexone as compared to the Placebo group. The STS has been implicated in the integration of information about others; for example, updating thoughts about others, tracking their outcomes and representing action and reactions of others[34–38]. In addition, the cluster in the STS was located within a mask that represents the overlap of reverse inference of the terms 'STS' and 'SOCIAL' generated from the NeuroSynth database[39] (threshold: $P(FDR) < 0.01$). The results of enhanced observational information processing fit well to an additional, exploratory PPI analyses of the left amygdala (reflecting the differences in responses towards the observational US between groups, see above). This analysis revealed enhanced connectivity between the amygdala and the visual association area (unpaired $t$-test, one-sided (Naltrexone $N = 22$, Placebo $N = 21$), extrastriate cortex, $P(uncorrected) < 0.001$; Supplementary Note 2; Supplementary Fig. 6), in the Naltrexone group as compared to Placebo.

These results showed that the blockade of opioid receptors during observational fear learning enhanced amygdala responses towards the observational US and sustained responses across trials in a region of the PAG that was functionally connected to the STS. These findings suggest enhanced processing of the observational US depended on an opioidergic innervated circuitry, which might be related to enhanced associative learning

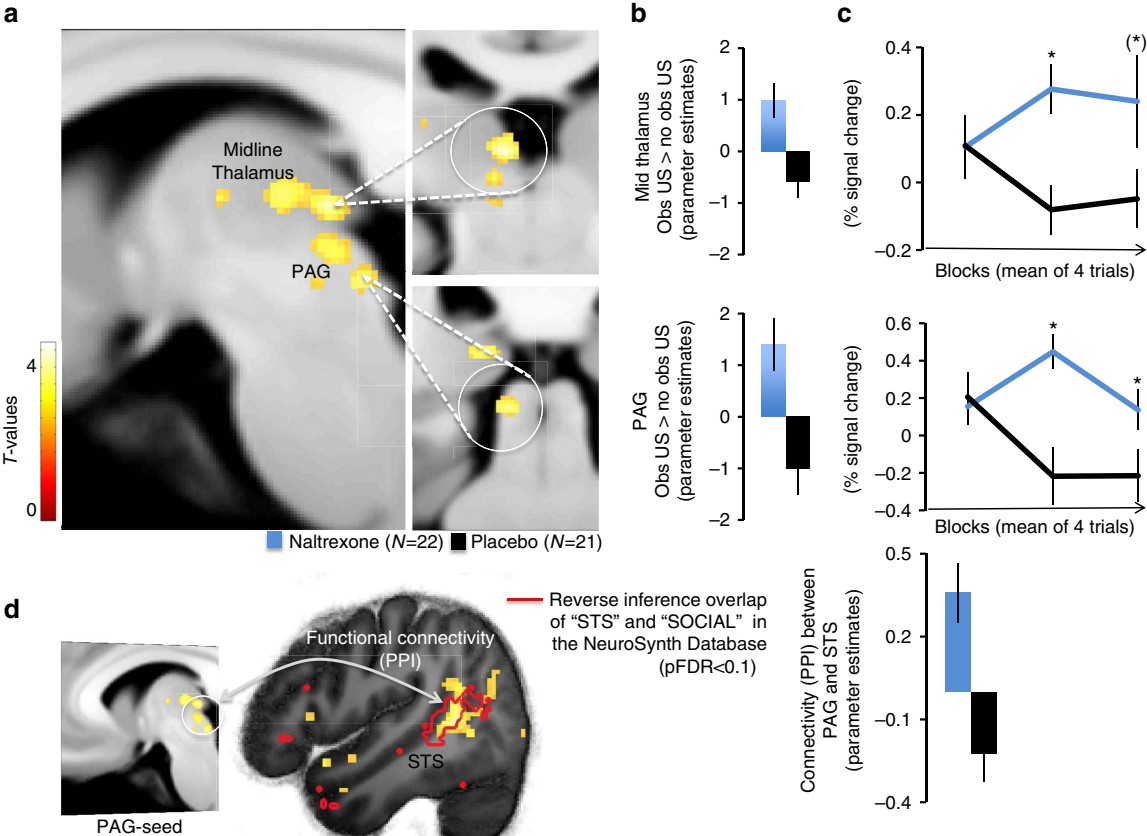

**Figure 3 | Temporal dynamics during observational threat learning.** (**a**) Brain BOLD responses to the observational US in the Naltrexone as compared to the Placebo group resampled to 1 cubic mm. (**b**) Blockade of opioid receptors in the Naltrexone group enhances responses towards the observational US in the PAG and midline thalamus as compared to Placebo controls. 'Obs US' refers to responses to the observational US and 'no obs US' to responses to CS+ outcomes that are not followed by the US. (**c**) Temporal dynamics of the midline thalamus (upper) and PAG (lower) indicates decreasing responses to the observational US as a function of learning in the Placebo group and sustained observational US responses in the Naltrexone group, see Supplementary Fig. 3 for specific response to obs US and no obs US trials. (**d**) PAG responses displayed an increased functional connectivity (PPI) within the superior temporal sulcus (STS) in the Naltrexone, as compared to Placebo, group (unpaired t-test, one-sided (Naltrexone $N = 22$, Placebo $N = 21$), $x,y,z$(MNI) = 46; − 50;8; $t = 3.94$, $P$(FWE, cluster) < 0.035). The cluster in the STS is located within a mask (red line) that represents the reverse inference of the terms STS and SOCIAL from the neurosynth data base with a threshold of $P$(FDR) < 0.01. The error-bars denote the standard error of the mean, and T-maps are superimposed on an average structural image with a threshold of $P$(uncorrected) < 0.01 for illustrative purposes. Asterisks indicate significant differences between groups, corrected for multiple comparisons (Bonferroni-Holmes).

and expression of conditioned fear responses. It is possible that this aversive learning circuit processes inputs from a social cognitive circuitry cantered on the STS that code information about the observed target individual (that is, observation of others in pain).

**Decoding opioid receptor function from threat expression.** Next we tested whether brain regions that were responsive to the observational US represented differences between groups during the expression of threat responses. We used a supervised machine-learning algorithm to classify individual pharmacological group status (that is, Naltrexone or Placebo) from CS+ responses during the immediate expression test with a kernel restricted to brain regions that were responsive to the observational US across groups ($P$(uncorrected ) < 0.001). We submitted the individual pattern of brain activity of all participants but one ($N − 1$), each classified as Naltrexone and Placebo group to train a Support Vector Machine (SVM). This trained classifier was then used to predict the participant that was left out ('leave one participant out' cross-validation). Training and test steps were repeated for each participant. Indeed, the SVM predicted the pharmacological challenge of the opioid receptor function (that is,

Naltrexone or Placebo) significantly above chance (permutation test with 1,000 permutations (Naltrexone $N = 22$, Placebo $N = 21$): $P = 0.01$, balanced accuracy: 72%). Computation of functional weights revealed the highest values for anterior temporal regions in proximity to the amygdala, including the bilateral anterior temporal gyrus and the left temporal pole. In addition, the right caudate and right thalamus contributed high weights, as well (see Fig. 4; Supplementary Table 9).

While we found no difference in threat expression between groups in the immediate test in the SCRs, this result might point towards a difference in brain activation patterns between groups during threat expression.

We then performed an exploratory univariate analysis to test whether the Naltrexone group showed enhanced haemodynamic responses during the immediate expression test in the amygdala. This was not the case: the comparison of averaged responses did not reveal any differences between groups. Next, we examined heamodynamic responses that linearly changed as a function of time, which should closely fit the psychophysiological expression of conditioned responses (SCR: linear stimulus by time interaction; $F(2,30) = 6.329$; $P = 0.007$, see Supplementary Table 1). Indeed, we found a linearly decreasing amygdala activity to the CS+ as compared to the CS− in the Placebo group, while

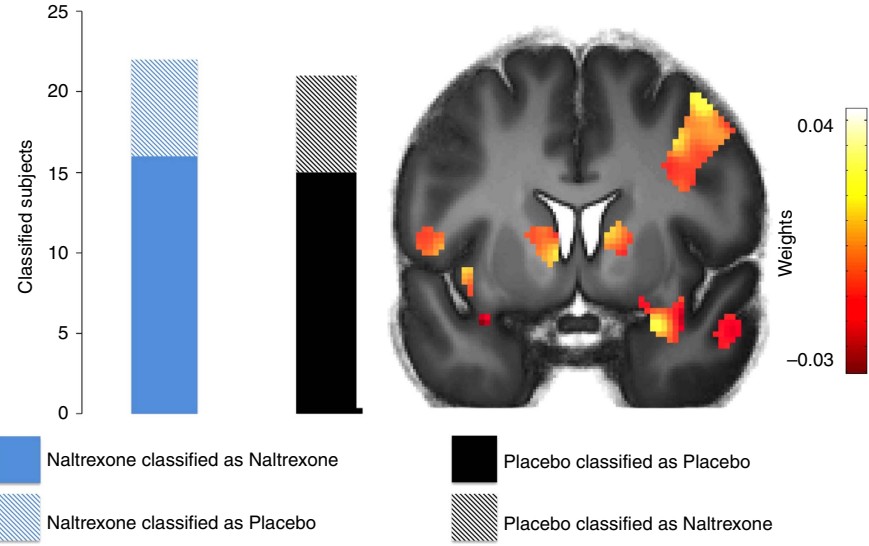

**Figure 4 | Classification of brain responses during immediate expression test.** Supervised machine-learning classification of haemodynamic pattern to the CS+ during the immediate expression test as Naltrexone or Placebo group. Highest values for classification were yielded in anterior temporal regions, including the bilateral anterior temporal gyrus and the left temporal pole, as well as the right caudate and right thalamus (see Supplementary Table 6 for a contribution of all regions). Please note that only regions responsive to the observational US were included in the training kernel. $T$-maps are superimposed on an average structural image with a threshold of $P$(uncorrected) $< 0.01$ for illustrative purposes.

amygdala responses in the same contrast in the Naltrexone group were sustained ($P$(FWE) $= 0.039$, see Supplementary Table 8).

## Discussion

In summary, our study shows that learning about threats by observing others is regulated by opioid receptors. Specifically, we found that the blockade of opioid receptors enhanced response to the other individual's distress, the so-called observational US, in the amygdala, midline thalamus and the PAG, leading to more persistent observational fear conditioning. The response pattern in the PAG displayed the characteristics of a teaching signal as described in experiments in rodents and humans during directly experienced painful events[27–29,32]. During observational threat learning, PAG responses towards the aversive outcome to the other were sustained through opioid receptor antagonism, and predictive of learned threat responses to the conditioned stimuli, CS. Moreover, this teaching signal in the PAG was functionally connected to responses in the STS, a region that has been implicated in the processing of outcomes to others, as well as attributions of mental states to these individuals[37]. The linkage between enhanced processing of the observational US and the expression of threat responses was underlined by the result derived from multivariate machine-learning techniques. Training of a support vector machine with brain activity in regions that process the observational US could successfully decode opioid receptor function from brain responses during the immediate expression of threat responses. Finally, the enhanced expression of threat responses 72 h later (drug-free) was predicted by amygdala activity towards the observational US in the Naltrexone, but not in the Placebo, group.

Our findings describe a novel neuropharmacological challenge of observational fear conditioning that is functional relevant for the later expression of threat responses, pointing towards associative learning mechanism that might not be so different from those that mediate direct learning of threats[40]. Studies in humans and animals have shown that social transmission of defensive responses against sources of danger is effective[18,21,23,41–43] and our study suggests that an opioidergic

circuitry enables the individual to flexibly adapt to harmful and dangerous situations based on social information, even in the absence of directly experienced harm.

The neuropharmacological function of opioid receptors in observational threat learning fits well within a framework of opioidergic learning circuitry that translates salient (aversive/painful) events into expectations[28,44]. Such a circuit involves subcortical structures as the amygdala, PAG and the medial thalamus, together with cortical regions, such as the medial prefrontal cortex[28,32]. Moreover, our finding that the temporal dynamics of PAG scales aversive learning from observation of others is consistent with a recent finding in animals, showing that prediction error coding in the PAG (and the amygdala), sets aversive memory strengths during learning from direct aversive experiences[32].

Importantly, while previous research has revealed that this neural circuit computes predictions of directly experienced outcomes, we suggest that opioidergic processes are involved in outcomes that are transmitted by social means. Our neuropharmacological results are in line with previous research showing that observed rewarding outcomes in others are computed and updated by similar neural mechanisms that process directly experienced outcomes across species[13,14,45–47]. Hence, this opioidergic circuit might code representations of aversive outcomes, derived through both direct and indirect experiences. Consistent with this reasoning, a previous study in rodents showed that Naltrexone enhanced second order conditioning; the association between a previously paired CS (learned representation of pain) and a neutral stimulus[48]. In addition, other (learned) representations/regulations of pain are evidently regulated by opioidergic transmission, such as placebo manipulations of pain[44,49–53]. Interestingly, even placebo manipulations that regulate vicarious pain when observing other individuals experiencing pain are sensitive to opioidergic transmission[54]. Thus, the opioid system seems to process real or vicarious pain in a similar way. However, it is important to note that although real and vicarious pain activations include several overlapping brain areas these activations do not necessarily reflect the same underlying mechanism[55]. Rather, it might be that

regions that are active during real and observed pain represent central hubs within networks that code aversive/salient stimuli that receive and process divergent inputs. In our study, the PAG was functionally connected with the STS, which might be one such (input-)region that has been implicated in the processing of social information. Hence, learning from aversive outcomes, regardless of their origin, might be processed by similar structures.

Our findings mirror previous research on fear states, showing that the amygdala, thalamus and medial prefrontal regions decrease their US signalling (to directly experienced USs) with increasing US expectancy during fear conditioning in humans[56]. In addition, our results fit to research on different fear/threat states in which the passive post-encounter state (activated when a threat is observed) is opioid dependent and involves central amygdala and ventrolateral PAG, while the active circa-strike state (initiated when there is an active attack) is dependent on non-opioid mechanisms and the dorsolateral PAG[5,57].

In addition to its involvement in the defense of physical harm, the opioidergic system in humans and other animals has been implicated in reward and motivation related to conspecifics. In particular, behavioural responses elicited by affiliation, social connection and rejection have been found to be mediated by endogenous ligands at the opioid receptor[58–60].

The results presented in this study suggest a novel pharmacological mechanism of social threat learning and therefore inherently bear some limitations. We cannot exclude that processes preventing habituation or extinction (in particular during the test stages), as well as increasing sensitization, might have contributed to our results. Future studies are needed to disentangle social threat learning mechanisms in more detail to refine the neuropharmacological model of social threat learning in humans. In addition, our results revealed co-activity in a neighbouring structures of the PAG, including the colliculus, which has been described as both an output and input region to the PAG[61]. Future research is warranted to employ high resolution of the brainstem function in observational learning to describe the contribution of the PAG and neighbouring regions in greater details.

However, connecting these findings with our results suggests that endogenous opioids regulate expectations and responses that arise from outcomes of both physical and social salience. Our study extends earlier findings by identifying an opioidergically regulated mechanism that translates indirect, social, experiences into own expectations and responses.

In summary, our study shows that endogenous opioids, which serve as potent analgesics against the direct experience of pain also code social threat learning from pain that is transmitted solely through observation. Social learning mechanisms might thus be using an opioidergic circuitry that is responsive to direct aversive experiences, to learn more efficiently through the experiences of others.

## Methods

**Participants.** A total of 43 healthy male adults (right-handed = participants who were free from self-reported life-time psychiatric or neurological disease and medication were recruited and participated in the study. The experiment was approved by the Regional Ethical Review Board in Stockholm (www.epn.se) All participants gave written informed consent and were paid 1000 SEK (approximately 120 USD) for their participation. There were no differences with regard to STAI trait anxiety ($P > 0.7$, $t < 1$), STAI state anxiety ($P > 0.2$, $t < 1.2$), or BEES ($P > 0.9$, $t < 1$) between participants in the Naltrexone or Placebo group.

**Pharmacological manipulation.** Forty-three participants were randomly assigned to the Naltrexone or Placebo group in a double-blind manner. Participants received the study medication (50 mg Naltrexone, or Vitamine E) after a negative test for drugs and being in fasting condition for least 60 min. Approximately 90 min after drug-intake (receptor blockade in the brain is approximately expected 60 min after

oral administration[62]), participants were placed in the fMRI environment to start the experiment.

**Stimuli and experimental timing and procedure.** Before starting the experimental task, participants were attached to SCR and shock electrodes with the instruction that they might receive a shock at any time during the whole time-course of the experiment. During observational fear conditioning, videos were presented that showed the demonstrator sitting in front of a computer screen watching two differently coloured squares (yellow and blue), serving as observational CSs. Observational CSs were presented for 9 s and each CS was presented 24 times in total. During in 50% of the observational CS + trials observational USs were delivered 6.5 s after CS onset. The demonstrator reacted to the shocks by slightly twitching the arm and blinking (resulting from an electric stimulation of the shock electrode that was visibly attached to the demonstrator's right wrist). Observational US delivery at CS + trials was pseudorandomly assigned (8 CS + trials were randomly paired with 4 USs). The demonstrator acted calmly while watching the presentations of the CS − . During the direct fear expression tests, the same CSs (blue and yellow coloured squares) were presented directly to the participants, again presented for 9 s, 12 presentation of each CS during the immediate test stage and 6 presentation of each CS during the long-term test stage. No directly presented CS was followed by a US. ITIs during observational and direct fear expression test were jittered between 4 to 7 s.

**Subjective ratings.** Participants were asked after the immediate expression stage for the amount of fear and US expectancy (assessing CS–US contingency) evoked by each CS. In addition, they were asked how unpleasant its is to observe the US and how unpleasant they think it is for demonstrator (see Supplementary Fig. 2 and Supplementary Note 1 for details).

**SCR analysis.** As in previous studies[63], skin conductance responses (SCRs) were scored as increase in skin conductance within 0.5 to 4.5 s after stimulus onset, range-corrected [(SCR/SCRmax) + 1] and logarithmized. Due to reduction in data quality in the MR-environment, SCRs of 10 participants (Placebo $N = 6$, Naltrexone $N = 4$) were excluded from the analysis during the observational learning and Immediate test stage. One participant (Placebo) had missing data in more than 50% of the trials and was excluded from the analysis of the Long-term test stage. Averaged SCRs of 4 trials were entered in a repeated measurement ANOVA with a within subject factors (CS-type; 2 levels) and pharmacological group as a between subject factor. In addition, the analyses of the immediate test stage included block (3 levels) as a factor. Our main focus of analyses examined CS discrimination (that is, CS + > CS − ) as an indicator of successful conditioning and expression of conditioned responses between groups (that is, Naltrexone > Placebo, see hypothesis in the main text).

**fMRI acquisition.** fMRI data were acquired using a 3 Tesla MR scanner (General Electrics 750) with an 8-channel head coil. Each functional image volume comprised 47 continuous axial slices (3 mm thick, 0.7 mm gap) that were acquired using a T2*-sensitive gradient echo-planar imaging (EPI) sequence [repetition time (TR): 2,870 ms; echo time (TE): 30 ms; flip angle: 90°; 2.3 × 2.3 mm in-plane resolution]. The first 5 volumes of each time series were discarded to account for T1 equilibrium effects. Pre-processing involved distortion correction of susceptibility-induced gradients of BOLD images through field maps, realignment, unwarping, co-registration and normalization to a sample-specific template using DARTEL and spatial smoothing (6 mm FWHM isotropic Gaussian kernel) within the 'Statistical parametric mapping' (SPM8, www.fil.ion.ucl.ac.uk/spm) software package. Further processing accounted for field inhomogeneity through acquired field maps, temporal high-pass filtering (cutoff 128 s) and correction for temporal auto-correlations using first-order autoregressive modelling. In addition, brainstem centred normalization (box dimensions: x: − 30 to 30; y: − 45 to 0 and z: − 50 to 18 mm) with a re-sliced resolution of 1 cubic mm and spatial smoothing with 4 mm FWHM isotropic Gaussian kernel was performed in order to identify the activity in these ROI localized in these small regions. A general linear model modelled each onset of the observational CS + and the observational CS − , each outcome of the CSs (observational US after CS +, no observational US after CS + and no observational US after CS − ) as well as each onset of the direct CS + and the CS − . All regressors were convolved with a canonical hemodynamic response function. Random effect analysis ('full factorial' model) on the group level was performed on the individual beta estimates of CS + outcomes (that is, observational US versus no observational US) during the observational fear conditioning. The preceding CS does not influence responses towards the observational US were, because only CS + outcomes were contrasted.

**Functional connectivity analysis.** PPI (as implemented in SPM8, see Supplementary Methods for details) was used to examine functional connectivity differences of PAG responses towards the observational US (observational US > no observational US) between groups. Extracted eigenvariates of the PAG peak voxel were used as the seed region, deconvolved and controlled for the PAG time-course and the onset regressor.

**Supervised machine learning classification.** Supervised machine-learning classification analysis was employed using SVM as implemented in the Pattern Recognition for Neuroimaging Toolbox for SPM8 (ref. 64), entering the individual beta estimates of CS+ response during the immediate expression test as inputs. The kernel was restricted to brain regions that were responsive to the observational US (observational US > no observational US, $P < 0.001$ uncorrected) and cross-validation of the classification was using the 'leave one participant out' method.

**Regions of interest.** The amygdala ROI was defined as a probabilistic anatomical mask (threshold 0.7)[65] and the PAG was defined as a sphere (4 mm) around a peak coordinate ($x: \pm 6$; $y: -34$; $z: -6$) from a previous study representing the conjunction of the experience and anticipation of direct pain[66]. In addition, we constructed a box representing the standard deviation around the meta-analytic average of 225 PGA responses[67] with the dimension $x: -4 (\pm 3)$, $y: -29 (\pm 5)$, $z: -12 (\pm 7)$; see Supplementary Fig. 7 and Supplementary Note 2. The sub-region of the midline thalamus ROI was defined as a probabilistic mask of functional connectivity, where the midline thalamus is connected with temporal and prefrontal regions} (threshold 0.7)[68].

$P$ values inside the ROI were corrected for multiple testing (small-volume correction, SVC) using family-wise error (FWE) correction. In all analysis an alpha level of 0.05 was adopted, but marginally significant results ($P < 0.10$) are also reported.

**Data availability.** We confirm that all relevant data are available from the authors. Neuroimaging data are available in a public repository at http://neurovault.org/ with the following accession code: /collections/YLMBGZKG/.

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

## Acknowledgements

This research was supported by the European Research Council (Independent Starting Grant 284366; Emotional Learning in Social Interaction to A.O.) and the Knut and Alice Wallenberg Foundation (KAW 2014.0237) to A.O., and the German Research Foundation (Research Stipend HA 7470/1-1 and SFB/TRR 58 to J.H.)

## Author contributions

J.H. and A.O. conceived the experiment with support from P.P., J.H. and J.Y. conducted the experiment, J.H. analysed the results. J.H. and A.O. drafted the manuscript and all authors contributed revisions.

## Additional information

**Competing interests:** The authors declare no competing financial interests.

**Publisher's note**: 

