## [Peer Review File · Nature Communications]

Reviewers' comments:

Reviewer #1 (Remarks to the Author):

Haaker et al report the results of an interesting experiment assessing effects of naltrexone on social threat learning in humans using a combination of pharmacology, fMRI, connectivity, and supervised machine-learning.

1. The key findings here are that naltrexone facilitated observational fear learning, and that this behavioral effect was related to observational US-related hemodynamic activity in PAG, midline thalamus and amygdala, and that PAG showed strong functional coupling with the STS during this task.

2. The use of the observational learning task is interesting and is an important source of the novelty of these results. The findings and conclusions are important confirmation of some of the key claims of the prediction error model opioid contributions to fear learning, proposed in the non-human literature by McNally as well as Fanselow. Moreover, they show quite neatly that this role in humans cannot just be linked to physical noxious stimulation and analgesia because the US here was observational

In addition, these data represent a very important extension of the work by Eippert et al. who have conducted similar studies, but for direct fear learning, in humans.

Moreover, because this preparation involved observation, not an explicitly painful US, they support the claims that the roles of opioids in aversive learning can not be reduced simply to pain modulation. For each of these reasons I think this manuscript is an interesting and important new contribution to the literature.

3. I struggled, in parts, to grasp features of the data and imaging

a. p. 2, the significance of the naloxone effect on long-term expression test SCRs depends on the use of a one-tailed test. This could be problematic, but does not need to be. I think the authors need to be frank about this: if they were predicting (as they were) an effect in that direction and not the other; whether the choice about one tailed was made in advance; and perhaps also noting that past research with naloxone and SCRs has yielded mixed effects but in the same participants yielded stronger effects in reaction time data.

b. I had much difficulty tracking and understanding the changes in df across the various experiments and this could be made clearer, especially given the exclusion of participants based on SCRs.

c. Were the participants excluded on the basis of SCRs excluded on any other measures or analyses?

d. The PAG and midline thalamus are very small regions in the human brain. I would appreciate more detail on how signals in these regions were localized.

e. The PPI analysis stills remains unclear to me. Did the authors test alternate seed locations?

f. The temporal coding analysis was important, because it shows a neat diminution of US

evoked signals across trials. However, I could not really see why the authors then relied on simple comparisons. Why not just compare linear trends?

g. Much of the previous work in this field has noted that PAG, thalamic, and amygdala hemodynamic responses in aversive learning tend to co-vary with medial prefrontal and, sometimes, more dorsolateral prefrontal, regions. It may be useful to comment on this, even if briefly, in the discussion.

4. The authors are to be commended for their scholarly and fair citations of the relevant animal and human neuroscience literature. I would just note that the work of Dunsmoor and colleagues on variations in shock US evoked hemodynamic signals in the human thalamus and medial prefrontal cortex are relevant here; they serve as useful comparison (as does the cited work or Eippert) between observed and actual aversive USs.

5. I do think it is important to state, in the main text, that the participants “were attached to SCR and shock electrodes” as is well stated in the Methods. This will help readers unfamiliar with observational and instructional fear learning better understand the nature of the task.

6. The authors emphasize that the pattern of hemodynamic responses here “display the characteristics of a teaching signal”. In effect, the US-related signal decreases as the US is less surprising or becomes more expected. I would modify this conclusion. These data show one characteristic (variations in the effectiveness of the US across conditioning), and the effect shown is important, but it could also be linked to actions as preventing habituation, causing sensitisation etc. Other characteristics – such as blocking, extinction etc – are needed to better link these changes to a teaching signal in aversive learning (e.g., Eippert, 2012). I am not suggesting that the authors complete such experiments, rather that there remain some alternate interpretations of this pattern and profile of signals.

Reviewer #2 (Remarks to the Author):

In this study by Haaker and colleagues, the authors examine how observational threat learning is coordinated and the role of the endogenous opioid system in this process. They show that blocking opioid receptors using systemic administration of naltrexone increases long-term observational threat learning as indexed by enhanced skin conductance responses to a CS+ compared with a CS-. Using fMRI they look at the brain correlates of this and find enhanced observational US processing in the amygdala, periaqueductal gray (PAG) and midline thalamus. Interestingly, when examining the temporal dynamics of the observational US response in PAG and thalamus they suggest that it reduces across the learning session and that this learning dependent reduction is blocked in the naltrexone treated group. Using cross-regional interaction analysis, they go on to show that the PAG response is correlated with activity in the superior temporal sulcus. Finally, using a decoding approach in areas that showed observational US responses they show that they can predict the drug treatment group based on CS+ responses.

While the idea is intriguing there are substantial analytic issues that make it difficult to draw any firm conclusions from the data. In addition, it seems that the authors are taking on a question (how the endogenous opioid system contributes to observational threat learning) that first requires a determination of what brain systems mediate observational threat learning under normal conditions. If these brain regions are only activated under conditions of opioid receptor blockade the conceptual advance is seriously undermined. I've outlined specific critiques below, but generally speaking I'm not enthusiastic about the paper as it now stands.

1) The behavioral SCR analysis presented in Fig. S1 and Fig. 2a is somewhat weak. Generally, the definition of 'learning' is vague and whether subjects are learning the CS+/observational US association is not apparent from the statistical approach the authors used. It seems that they want to say that learning represents a significantly higher SCR to the CS+ compared with the CS-. If so then they need to analyze and report that statistical analysis throughout. For the short term Immediate expression test they do show this data (Fig. S1). However, here they only report an F test and say that they found no differences between groups (they cite a $p > 0.2$, but I can't tell what kind of statistic was used for this test or what is actually analyzed). What they need is an ANOVA followed by post-hoc analyses to determine whether there is a significant difference between CS+ and CS- in the placebo and naltrexone conditions. From Fig. S1 it looks like there is a difference between CS+ and CS- only in the placebo conditions undermining their claim that there is learning in both conditions. If they want to define learning as an increase in CS+ SCR over some pre-learning baseline period then they need to state that in the text and perform appropriate statistical tests.

2) The same issue as in point 1 above arises in Figure 2a where they claim that there is more learning in the naltrexone compared with the placebo control. Here though they don't report the raw CS+ and CS- SCR values, but rather directly compare the contrast across the two conditions. This is problematic because we again don't know how learning is defined and it is difficult to understand what the data represent without the same comparison they perform for the Immediate expression test. They should pick one analysis, define clearly what learning is and test their hypotheses according to that measure. Showing the non contrasted CS+ and CS- for the data in Fig. 2a is important regardless of what they finally decide to do.

3) Similar to the behavioral analysis, for the amygdala, PAG and thalamic responses to the observational US (Figs. 2c, 3b-c), while it is clear that there are differences between the naltrexone and placebo group, it is not clear whether there is a significant response to the observational US in the placebo group compared to some baseline. This is an important issue as without a significant response in somewhat normal conditions (placebo) it is hard to say that these brain regions process observational USs and suggests that what they are seeing in the naltrexone is an artifact of the drug treatment itself.

4) Related to the above (point 4), it is also unclear whether there is a significant training induced reduction in the observational US response in PAG and thalamus in the placebo group (Fig. 3b-c). Again, this is important to establish that they are seeing some kind of observational US prediction error response in these regions as they suggest.

5) In the cross-regional interaction analyses in Fig. 3d the authors say that "PAG responses displayed an increased functional connectivity (PPI) with the ...STS". From my reading of the paper and Methods it is not clear what responses they are analyzing here. Observational US? CS+?

6) In the decoding analysis presented in Fig. 4, they use their supervised machine learning approach to decode treatment group from the CS+ responses during the immediate expression test. As they saw no behavioral differences (but see point 1) at this timepoint it is not clear why they are looking here rather than at the long term test timepoint. This should at least be discussed if not extended to the later timepoint.

7) Also related to the decoding analysis, the authors should examine individual brain regions for classification analysis (in conjunction with the whole network analysis that they use now) to determine which brain regions are most important for the decoding.

8) Related more generally to the effect of naltrexone on brain responding, it is possible that the drug is changing the brain in some way that it appears more active than in placebo conditions (ex. Changes in resting activity or general stimulus evoked activity). This could explain the effects US processing, cross-region coupling and drug treatment decoding they observe. The authors should try to address this in some way.

Minor Points

1) They reference Table S1 on pg 2 in support of the idea that the amygdala is activated by observational USs, but from my reading it looks like Table S1 only deals with PAG responses.

2) The figure legend for Fig. 3 discusses a panel 'e' which is not in the figure.

Reviewer #3 (Remarks to the Author):

In this manuscript, the authors examine the hypothesis that opioidergic neural circuits shape prediction error during observational threat conditioning in humans. The results show that the opioid antagonist naltrexone, given prior to observational fear learning, enhances amygdala activity to the observed US, and also produces a correlation between amygdala activity to the US during learning and the degree of long-term memory measured 3 days later. The authors also report clusters of activity evoked by the observed US in the midline thalamus and PAG that are greater for the naltrexone group compared to the placebo controls.

The findings of this study are largely confirmatory. That is, they show that neural fear circuits identified in animal studies are also important in humans. The major contribution here is showing that opioids are important for observational fear memory in humans. However, the role of endogenous opioids seems to eliminate, rather than "limit" (as claimed by the authors), observational fear because most subjects in the placebo control group did

not have long-term observational fear memory. Overall, I find the question to be an interesting one, but I'm not sure that the findings reach the novelty level required for Nature Communications.

1) The authors note that the groups, on average, show evidence of learning (Figure S1). Yet, it seems clear from Figure 1b that a substantial number of participants in both the placebo and naltrexone groups must not have learned the task. In this graph, a negative difference between the CS+ and CS- indicates that a subject had a greater SCR for the CS- than the CS+. It is not clear to me why these subjects who did not learn the task would be included in any analysis. Also, it appears that, on average, participants in the placebo group (Figure 1a) didn't show long-term memory, because the average SCR difference score was negative. How can the authors claim that opioid receptor blockade "enhanced observational fear learning" if there was no observational fear learning in the controls (placebo group)?

2) It is not clear from the text whether the analyses restricted to the "observed US" period only used data from the 12 trials in which the observed US was actually presented. Please clarify.

3) Figure 3b It isn't clear what the "no obs US" trials are. Are these the CS+ trials without any US presented? Are they the CS- trials? This type of trial-by-trial analysis should be shown for the amygdala as well.

4) The Figure 4 legends references error bars, but no error bars are depicted in the graphs.

Reviewer #1 (Remarks to the Author):

Haaker et al report the results of an interesting experiment assessing effects of naltrexone on social threat learning in humans using a combination of pharmacology, fMRI, , connectivity, and supervised machine-learning.

1. The key findings here are that naltrexone facilitated observational fear learning, and that this behavioral effect was related to observational US-related hemodynamic activity in PAG, midline thalamus and amygdala, and that PAG showed strong functional coupling with the STS during this task.

2. The use of the observational learning task is interesting and is an important source of the novelty of these results. The findings and conclusions are important confirmation of some of the key claims of the prediction error model opioid contributions to fear learning, proposed in the non-human literature by McNally as well as Fanselow. Moreover, they show quite neatly that this role in humans cannot just be linked to physical noxious stimulation and analgesia because the US here was observational

In addition, these data represent a very important extension of the work by Eippert et al. who have conducted similar studies, but for direct fear learning, in humans.

Moreover, because this preparation involved observation, not an explicitly painful US, they support the claims that the roles of opioids in aversive learning can not be reduced simply to pain modulation. For each of these reasons I think this manuscript is an interesting and important new contribution to the literature.

Response:

We are delighted that the reviewer shares our enthusiasm about the current study, and that he/she considers it a “..very important extension..” of existing work, and an “..important new contribution to the literature”. Furthermore, we are grateful for his/her suggestions for additional analyses that extended our findings.

In order to facilitate the review process, we have highlighted in yellow the changes made to the main text, and included these text excerpts in the reply here below.

3. I struggled, in parts, to grasp features of the data and imaging

a. p. 2, the significance of the naloxone effect on long-term expression test SCRs depends on the use of a one-tailed test. This could be problematic, but does not need to be. I think the authors need to be frank about this: if they were predicting (as they were) an effect in that direction and not the other; whether the choice about one tailed was

made in advance; and perhaps also noting that past research with naloxone and SCRs has yielded mixed effects but in the same participants yielded stronger effects in reaction time data.

Response:

We thank the reviewer for bringing our attention to this. As the reviewer states, we did hypothesize a higher long-term expression of conditioned responses in the Naltrexone as compared to the Placebo group (as stated in the main text, page 1): "...we conjectured that opioid receptor blockade would enhance neural signalling (fMRI) of the observational US in the PAG and amygdala, leading to stronger long-term expression of conditioned responses."

We also agree that our analysis strategy can be stated even more clearly, by adding the contrast in which the groups were expected to differ.

Now stated in the methods: "Our main focus of analyses examined CS discrimination (i.e. CS+>CS-) as an indicator of successful conditioning and expression of conditioned responses between groups (i.e. Naltrexone > Placebo, see hypothesis in the main text)"

Now stated on page 2: "Then, we tested our hypothesis if blockade of opioid receptors during observational fear learning enhanced long-term expression of threat responses (i.e. enhanced CS discrimination)."

Additionally, we have added the results of the ANOVA including both, CS+ and CS- responses, before reporting the t-test in order to bolster the validity of our analysis (page 2): "Indeed, the Naltrexone group expressed greater threat responses as compared to the Placebo controls in the drug-free long-term expression test [stimulus by group interaction: $F(1,40)=3.713$; $p=0.061$; $\eta^2=0.085$; t-test one-tailed of CS discrimination between groups, $t(40)=1.927$; $p=0.030$, see figure 2a and supplementary figure S1 b for CS specific responses]."

b. I had much difficulty tracking and understanding the changes in df across the various experiments and this could me made clearer, especially given the exclusion of participants based on SCRs.

Response:

We apologize that we have not been clear about the changing number of participants in our analyses. While all participants were included in the fMRI analyses, we had to exclude some participants in our SCRs analyses, due to reduced SCR-data quality.

In the methods we stated that (page 8): "Due to reduction in data quality in the MR-environment, SCRs of 10 participants (Placebo N=6, Naltrexone N=4) were excluded from the analysis during the observational learning and Immediate test stage."

Additionally, at the long-term test stage 1 Participant (Placebo) had a large amount (>50%) of missing data, and was classified as a non-responder.

The reviewer is correct in that we did not describe the exclusion for the long-term test in the main text. We changed that (and adjusted the df, see responses above) and hope that our revised manuscript is clearer now.

We added on page 2: "... analysis of skin conductance responses (SCRs, see methods for exclusion criteria)" And on page 8: "One participant (Placebo) had missing data in more than 50% of the trials and was excluded from the analysis of the Long-term test stage."

c. Were the participants excluded on the basis of SCRs excluded on any other measures or analyses?

Response:

The participants with lower SCR data quality were excluded from the SCR analyses only. The exclusion criteria applied for the fMRI data were based on the movement parameters, as well as on artefacts in the acquisition. Fortunately, no fMRI data had to be excluded.

d. The PAG and midline thalamus are very small regions in the human brain. I would appreciate more detail on how signals in these regions were localized.

Response:

We have followed the reviewer's suggestion and added more information about the definitions of the ROIs in the main text (Page 9):

"The amygdala ROI was defined as a probabilistic anatomical mask (threshold 0.7)⁶⁰ and the PAG was defined as a sphere (4mm) around a peak coordinate (x: +/- 6; y: -34; z: -6) from a previous study representing the conjunction of the experience and anticipation of direct pain⁶¹. The sub- region of the midline thalamus ROI was defined as a probabilistic mask of functional connectivity, where the midline thalamus is connected with temporal and prefrontal regions (threshold 0.7)⁶²"

Additionally, we applied a brain-stem centred normalization with a re-sliced resolution of 1 cubic mm in order to identify the activity in these ROIs localized in these small regions. We have added this information to the methods in the methods in the main text (Page 8): "Additionally, brain-stem centred normalization (box dimensions: x: -30 to 30; y: -45 to 0 and z: -50 to 18 mm) with a re-sliced resolution of 1 cubic mm and spatial smoothing with 4 mm FWHM isotropic Gaussian kernel was performed in order to identify the activity in these ROI localized in these small regions."

Regional location was then compared to MRI based human atlases. See figure below for illustrative examples.

Medial Thalamus

PAG

e. The PPI analysis stills remains unclear to me. Did the authors test alternate seed locations?

Response:

We apologize for being unclear in the description of the PPI. The PPI analyses used the seed-voxel displaying the differences in observational US responses (observational US > no observational US) in the PAG (since the time-course was the most interesting) between groups. Each individual time-course was then deconvolved and multiplied with the condition specific onsets of the observational US > no observational US contrast. This resulting psychophysiological interaction was then entered into a GLM, which controlled for the PAG time-course and the onset regressor. The beta-estimates of this GLM were then compared between groups. In order to enhance transparency, we have included a short method description of the PPI in the main text (Page 8):

“Psycho-physiological interaction (PPI, as implemented in SPM8, see supplementary methods for details) was used to examine functional connectivity differences of PAG responses towards the observational US (observational US > no observational US) between groups. Extracted eigenvariates of the PAG peak voxel were used as the seed region, deconvolved and controlled for the PAG time-course and the onset regressor.”

The reviewer might additionally wonder if we examined the connectivity between other

structures, such as the midline thalamus or the amygdala. We have not initially done that, because the time-course of the PAG stood out from our results and we wanted to explore structures that showed a similar time-course. However, inspired by the reviewers' suggestion, we explored functional connectivity between groups with a seed region in the midline thalamus and the amygdala (reflecting the difference between groups).

Interestingly, we found that the Naltrexone group (as compared to Placebo) showed higher functional connectivity between the amygdala and the extrastriate cortex in the visual association area (Brodmann area 19; x;y;z:-28;-90;2); $t=3.73$; $p(\text{uncorrected})<0.001$; see figure below). This complements our results of higher connectivity between the PAG and the STS, as reported the main text, suggesting that the Naltrexone group shows enhanced processing of observed information during the observational US. We added these results to the supplement (figure S6) and thank the reviewer for the suggestion to explore alternative seed regions. Moreover, we refer to these results in the main manuscript (Page 5): "The results of enhanced observational information processing fit well to an additional, exploratory PPI analyses of the left amygdala (reflecting the differences in responses towards the observational US between groups, see above). This analysis revealed enhanced connectivity between the amygdala and the visual association area (extrastriate cortex, $p(\text{uncorrected})<0.001$; see **supplementary results & figure S6**), in the Naltrexone group as compared to Placebo."

We did not find any group differences in connectivity using the medial thalamus as a seed region. Interestingly, however, we found that both, Placebo and Naltrexone group displayed connectivity between the midline thalamus and the dorsal medial PFC (x;y;z:3;22;53; $t=4.48$; $p(\text{uncorrected})<0.001$).

f. The temporal coding analysis was important, because it shows a neat diminution of US evoked signals across trials. However, I could not really see why the authors then relied on simple comparisons. Why not just compare linear trends?

Response:

We used an ANOVA that included block number as a factor, enabling us to compare linear or quadratic changes over blocks between groups. This was then followed up by simple comparisons as post-hoc test. This might not have been clearly stated in the main text, which we have now changed. We have also added the information about the type of temporal difference (Page 5): “Block by group interaction: $F(2,82)=4.88;p=0.011$, quadratic change over blocks $F(1,41)=7.15;p=0.011$; post-hoc two-tailed t-test} block 2: $p=0.002$ and block 3: $p=0.029$ ”

Interestingly, connecting these findings with the fMRI analyses reveals a similar result. Modelling an exponential decrease of fMRI responses towards the observational US (obs US > no obs US) revealed activity within the PAG for the Placebo group (but not the Naltrexone group), overlapping with the activity reported in our manuscript (4mm sphere; -8;-30;-10; $t=3.17$, $p(\text{SVC})=0.015$). This results confirms our initial analyses.

Related to this, we have calculated a first level that models simplified prediction error responses, defined as the deviation between the outcome and the expected outcome, to test if the PAG follows such a time-course. This Prediction error was modelled as the absolute difference between the observed outcome of CS+ trials (observational US = 1, no obs US = 0) and the sum of previous outcomes divided through the trial-numbers (i.e. average of outcomes of previous trials). The prediction error term was added as a parametric modulator of CS+ outcomes (controlling for the general outcome, i.e. obs US and no obs US).

A one sample t-test of activity in the Placebo groups revealed significant activity in the PAG (see figure below) reflecting the time-course of the prediction error (overlap with PAG activity in the manuscript x;y;z: -8;-30;-10; $t=3.77$; $p(\text{SVC}):0.030$; 4mm sphere). Interestingly, also other regions in the medial thalamus, medial PFC, and the amygdala followed this time-course, which is in accordance with previous research on neural correlates of Prediction errors in humans and animals.

Importantly, this prediction-error related coding in the PAG in the Placebo group was stronger as compared to the Naltrexone group, which did not follow a Prediction error related time-course (x;y;z: -8;-30;-10; $t=3.21$; $p(\text{SVC}):0.020$, see figure below).

We have added both analyses (exponential time-course and prediction error) to the revised supplement (see “**Additional temporal modelling of PAG responses towards the observational US**”). While our analyses in the manuscript constitute a rather simple approach, we believe that future studies that are explicitly designed to model prediction errors in the PAG might profit from our exploratory results.

g. Much of the previous work in this field has noted that PAG, thalamic, and amygdala hemodynamic responses in aversive learning tend to co-vary with medial prefrontal and, sometimes, more dorsolateral prefrontal, regions. It may be useful to comment on this, even if briefly, in the discussion.

Response:

We agree with the reviewer. In fact, on an uncorrected level, we found that medial prefrontal regions responded higher in the Naltrexone, as compared to the Placebo, group towards the observational US (i.e. co-activation with the PAG, thalamus and amygdala). Even though this was observed on a high uncorrected threshold, we have added this information in the main results (Page 3) and to table S 6: We contrasted hemodynamic activity towards the observational US between groups (Naltrexone > Placebo), which revealed higher responses to the observational US in the Naltrexone as compared to the Placebo group in the PAG (left x,y,z (MNI)=-8;-32;- 8; t=3.15;p(SVC)=0.016, see figure 3a,b). Additionally, we found a cluster in the midline thalamus (left x,y,z (MNI)=-6;-26;0; t=3.07; p(SVC)=0.038, see figure 3 a,b), as well as in the amygdala (as indicated by the previous analysis) and medial prefrontal areas (see table S6).”

Moreover, as suggested by the reviewer, we now comment on this in the discussion (Page 7): “Such a circuit involves subcortical structures as the amygdala, PAG and the medial thalamus, together with cortical regions, such as the medial prefrontal cortex²⁶.”

4. The authors are to be commended for their scholarly and fair citations of the relevant animal and human neuroscience literature. I would just note that the work of Dunsmoor and colleagues on variations in shock US evoked hemodynamic signals in the human thalamus and medial prefrontal cortex are relevant here; they serve as useful comparison (as does the cited work of Eippert) between observed and actual aversive USs.

Response:

We thank the reviewer for his/her valuable tip and have added a sentence about these findings, as well as a reference to the work by Dunsmoor and colleagues, to the discussion (Page 7):

“Our findings mirror previous research on fear states, showing that the amygdala, thalamus and medial prefrontal regions decrease their US signaling (to directly experienced USs) with increasing expectancy during fear conditioning in humans⁵³”.

5. I do think it is important to state, in the main text, that the participants “were attached to SCR and shock electrodes” as is well stated in the Methods. This will help readers unfamiliar with observational and instructional fear learning better understand the nature of the task.

Response:

We agree with the reviewer that this information should be added to the main text to facilitate the understanding in readers that are not familiar with this paradigm.

We have added this sentence to the revised manuscript on page 1:

“Participants were attached to SCR and shock electrodes during all stages”

6. The authors emphasize that the pattern of hemodynamic responses here “display the characteristics of a teaching signal”. In effect, the US-related signal decreases as the US is less surprising or becomes more expected. I would modify this conclusion. These data show one characteristic (variations in the effectiveness of the US across conditioning), and the effect shown is important, but it could also be linked to actions as preventing habituation, causing sensitisation etc. Other characteristics – such as blocking, extinction etc – are needed to better link these changes to a teaching signal in aversive learning (e.g., Eippert, 2012). I am not suggesting that the authors complete such

experiments, rather than there remain some alternate interpretations of this pattern and profile of signals.

Response:

We agree with the reviewer that the discussion of additional processes is important, in particular since this is the first study that targets the pharmacological mechanisms of social threat learning in humans. Along the reviewer's suggestions, we have added the following to our revised discussion (See page 7):

“The results presented in this study suggest a novel pharmacological mechanism of social threat learning and therefore inherently bear some limitations. We cannot exclude that processes preventing habituation or extinction (in particular during the test stages), as well as increasing sensitisation, might have contributed to our results. Future studies are needed to disentangle social threat learning mechanisms in more detail to refine the neuropharmacological model of social threat learning in humans.”

Reviewer #2 (Remarks to the Author):

In this study by Haaker and colleagues, the authors examine how observational threat learning is coordinated and the role of the endogenous opioid system in this process. They show that blocking opioid receptors using systemic administration of naltrexone increases long-term observational threat learning as indexed by enhanced skin conductance responses to a CS+ compared with a CS-. Using fMRI they look at the brain correlates of this and find enhanced observational US processing in the amygdala, periaqueductal gray (PAG) and midline thalamus. Interestingly, when examining the temporal dynamics of the observational US response in PAG and thalamus they suggest that it reduces across the learning session and that this learning dependent reduction is blocked in the naltrexone treated group. Using cross-regional interaction analysis, they go on to show that the PAG response is correlated with activity in the superior temporal sulcus. Finally, using a decoding approach in areas that showed observational US responses they show that they can predict the drug treatment group based on CS+ responses.

While the idea is intriguing there are substantial analytic issues that make it difficult to draw any firm conclusions from the data. In addition, it seems that the authors are taking on a question (how the endogenous opioid system contributes to observational threat learning) that first requires a determination of what brain systems mediate observational threat learning under normal conditions. If these brain regions are only activated under conditions of opioid receptor blockade the conceptual advance is seriously undermined. I've outlined specific critiques below, but generally speaking I'm not enthusiastic about

the paper as it now stands.

Responses: We were happy to learn that the reviewer found the idea “intriguing” although he/she noted several analytic issues. We take these concerns seriously and therefore provide clarifications of our existing analytic strategy, as well as several additional analyses that support our findings of brain activity involved in observational fear conditioning under normal conditions (i.e. responses in the Placebo group), and how responses in these regions are changed after blockade of opioid receptors.

We thank the reviewer for his/her comments that we address below.

In order to facilitate the review process, we have highlighted in yellow the changes made to the main text, and included these text excerpts in the reply here below.

1) The behavioral SCR analysis presented in Fig. S1 and Fig. 2a is somewhat weak. Generally, the definition of ‘learning’ is vague and whether subjects are learning the CS+/observational US association is not apparent from the statistical approach the authors used. It seems that they want to say that learning represents a significantly higher SCR to the CS+ compared with the CS-. If so then they need to analyze and report that statistical analysis throughout. For the short term Immediate expression test they do show this data (Fig. S1). However, here they only report an F test and say that they found no differences between groups (they cite a $p > 0.2$, but I can’t tell what kind of statistic was used for this test or what is actually analyzed). What they need is an ANOVA followed by post-hoc analyses to determine whether there is a significant difference between CS+ and CS- in the placebo and naltrexone conditions. From Fig. S1 it looks like there is a difference between CS+ and CS- only in the placebo conditions undermining their claim that there is learning in both conditions. If they want to define learning as an increase in CS+ SCR over some pre-learning baseline period then they need to state that in the text and perform appropriate statistical tests.

Response:

We apologize for our rather compressed presentation of the SCR results, and have now followed the reviewer’s suggestion to include more statistical details of the SCR analyses. The reviewer is correct in assuming that our primary interest is successful conditioning indexed by the discrimination between skin conductance responses, SCR, to the CSs (i.e. CS+>CS-). We agree with the reviewer that we could have stated and performed this strategy more stringently. Therefore, in the revised manuscript, we have now included the statistical strategy to the description of the SCR analyses and followed this strategy more stringently in the analyses. We have added the following to the methods (Page 8):

“Averaged SCRs of 4 trials were entered in a repeated measurement ANOVA with a within subject factors (CS-type; 2 levels) and pharmacological group as a between subject factor. Additionally, the analyses of the immediate test-stage included block (3 levels) as a factor. Our main focus of analyses examined CS discrimination (i.e. CS+>CS-) as an indicator of successful conditioning and expression of conditioned responses between groups (i.e. Naltrexone > Placebo, see hypothesis in the main text).”

We follow this strategy and now report the results in full depth (see supplementary table 1 and 2 with all results for the immediate and long-term test stage), including the separate test of CSs discrimination between groups during the immediate test stage as suggested by the reviewer (page 1):

“...indicated successful observational fear conditioning, i.e. CS discrimination, reflected as enhanced SCRs to the CS+ as compared to the CS- [Main effect of stimulus: $F(1,31)=5.215$; $p=0.029$; $\eta^2=0.144$; pair-wise comparison, CS+>CS-: $p=0.029$; see figure S1]. We found no differences between groups (Main effect of group or factor interaction with group: $p>0.6$; CS discrimination, i.e. CS+ > CS-, between groups did not differ: one-sided unpaired t-test, $t(31)<1$; $p=0.319$; see table S1) during the immediate expression test, consistent with subtle effects of Naltrexone on the expression of conditioned fear during direct conditioning⁶.”

Additionally, a separate ANOVA for the SCRs in the Placebo group only, revealed a trend-wise effect for the higher responses towards the CS+ as compared to the CS- throughout the immediate test-stage [$F(1,14)=3.46$; $p=0.084$], reflecting expression of conditioned responses (albeit reduced power).

Moreover, we followed the reviewer’s suggestion and now report the results of the repeated measures ANOVA for the long-term test SCR responses in the main text, and table S2, accompanied by CS specific bar graphs of the SCR responses. Page 1:

“Then, we tested our hypothesis if blockade of opioid receptors during observational fear learning enhanced long-term expression of threat responses (i.e. enhanced CS discrimination). Indeed, the Naltrexone group expressed greater threat responses as compared to the Placebo controls in the drug-free long-term expression test [stimulus by group interaction: $F(1,40)=3.713$; $p=0.061$; $\eta^2=0.085$; t-test one-tailed of CS discrimination between groups, $t(40)=1.927$; $p=0.030$, see figure 2a and table S2, as well as figure S1 b for CS specific responses].”

2) The same issue as in point 1 above arises in Figure 2a where they claim that there is more learning in the naltrexone compared with the placebo control. Here though they don’t report the raw CS+ and CS- SCR values, but rather directly compare the contrast across the two conditions. This is problematic because we again don’t know how learning is defined and it is difficult to understand what the data represent without the same comparison they perform for the Immediate expression test. They should pick one

analysis, define clearly what learning is and test their hypotheses according to that measure. Showing the non contrasted CS+ and CS- for the data in Fig. 2a is important regardless of what they finally decide to do.

Response: We agree with the reviewer, and to complement our additional results reporting the ANOVA for the long-term test (see responses above), we have added a plot of the CS specific response (see bar graph **b**) below) to the supplement. We agree with the reviewer that this plot provides substantial information and apologize that it was accidentally left out in the previous version. Furthermore, in the caption of figure 2, we direct the attention of the reader to the CS specific responses in the supplement (Figure 2 caption): “Fig 2 (a) Individuals receiving Naltrexone during observational threat learning showed enhanced conditioned fear responses (SCR, CS+>CS-; see **figure S1 for CS specific responses**) in the drug-free Long-term test.”

Interestingly, the Placebo group did not differentiate on average between the CS+ and the CS-, but both groups showed descriptively higher responses to the CS+ as compared to the CS- at the first trial (see figure below). This might suggest that both groups initially retrieved the CS-US association, yet this association was more persistent in the Naltrexone group.

Accordingly, we made the reported summary of our findings more specific (Page 6): “We found that the blockade of opioid receptors enhanced response to the other’s distress [...] leading to **more persistent** observational fear conditioning.”

3) Similar to the behavioral analysis, for the amygdala, PAG and thalamic responses to the observational US (Figs. 2c, 3b-c), while it is clear that there are differences between the naltrexone and placebo group, it is not clear whether there is a significant response to the observational US in the placebo group compared to some baseline. This is an important issue as without a significant response in somewhat normal conditions (placebo) it is hard to say that these brain regions process observational USs and suggests that what they are seeing in the naltrexone is an artifact of the drug treatment itself.

Response:

We think that the reviewer addresses an important point here and as we initially stated, we share his/her opinion that the comparison between the Naltrexone and Placebo group should reflect a “normal” mechanism in the Placebo group that is altered in the Naltrexone group. In general, the responses to the observational US are all contrasted responses to a “baseline”, i.e CS+ trials where no observational US was administered (obs US > no obs US) stated on page 2): “For that purpose, we contrasted responses across both groups towards the observational US (occurring at the end of 50% of the CS+ trials, (termed obs US) with responses during the same time-point to CS+ trials not followed by the US (termed no obs US), which controls for the influence of the preceding CS on observational US responses. This analytic approach, which has previously been used to study US responses during Pavlovian fear conditioning²⁴, will be employed in all of the following analyses.”

The first fMRI analysis in our manuscript follows the reviewer’s suggestion and does not contrast responses between groups, but examines brain regions that were responsive in the contrast observational US > no observational US across groups. This analysis revealed brain regions that were active in the Placebo as well as the Naltrexone group. One of these regions, which were defined as a-priori defined regions of interest, was the amygdala (bilateral, corrected for all independent voxel in the brain). In order to clarify

our approach, we now state in the revised manuscript (Page 2):

“We then tested if the amygdala was responsive to the observational US **across both groups** and if these responses were enhanced in the Naltrexone group. For that purpose, we contrasted responses **across both groups** towards the observational...”

In order to address the reviewer’s concern, we additionally examined responses towards the observational US (obs US > no obs US) in the Placebo group only and found significant responses in the amygdala (right: x,y,z (NMI): 20,- 8,-15; t=4.49; p(SVC, Amygdala ROI)=0.002; left: x,y,z (NMI): -22,- 8,-16; t=3.5; p(SVC, Amygdala ROI)=0.032, see figure below, right insert). This finding mirrors previous findings in humans (e.g. Olsson et al. 2007) and rodents (Jeon et al. 2010), showing that the amygdala is responsive to the observational US.

However, we understand that the reviewer received only limited information from figure 2c, since only contrast estimates (obs US > no obs US) were plotted there. Therefore, we have now added the plot of the singular responses towards observational US and no observational US trials in the supplement (supplementary figure 3, which is mentioned in the figure caption of figure 2: “(see **figure S3** for specific response to Obs US and no obs US trials).” See figure below.

Accordingly, we changed the simple comparison of contrast estimates in the Amygdala to a 2x2 ANOVA (obs US/no obs US trials ; Naltrexone/Placebo), Page 3:

“The comparison of averaged parameter estimates in the bilateral amygdala ROI between groups revealed **a main effect of stimulus (F(1,41)=59.795, p<0.001; obs US > no obs US), as well as a stimulus* group interaction (F(1,41)=5.967, p<0.019), representing a higher differential response to the observational US (obs US > no obs US) in the Naltrexone, as compared to the Placebo group, see figure 2c and figure S3 for condition-specific responses)”**

We think that our analytic approach, taken together with the additional information and previous studies in humans and rodents, suggest that the amygdala responds to the

observational US are a “normal” function that is altered through blockade of endogenous opioids. We have added this conclusion on page 2:

“Taken together, these results indicate that the amygdala is responsive towards the observational US during observational fear learning under normal conditions, and that this responsivity can be enhanced by the blockade of opioid receptors...”

With regard to the responses in the PAG and the midline thalamus, we initially followed a different analysis-strategy in the manuscript. We explicitly contrasted responses between observational US and no obs US trials that were higher in the Naltrexone as compared to the Placebo group. The reviewer is right that these responses might then represent an artefact in the Naltrexone group. However, figure 3c shows that the contrast estimates (obs US > no obs US) in the Placebo and Naltrexone group are comparable within the first block. We expected this time-course of PAG and thalamic responses in the Placebo group, i.e. responses towards the observational US in the beginning of the experiment that are decreasing throughout the experiment. We explicitly stated this strategy initially on page 5: “[...]studies in both species have highlighted the importance of the PAG for learning to predict aversive events, and have shown that responses towards directly experienced decrease over the time-course of learning^{25–27}. Based on these findings, we tested if the (on average) reduced responses in the PAG in the Placebo as compared to Naltrexone group resulted from a difference in temporal dynamics.”.

However, we understand that condition specific plots for the estimates representing observational US and no observational US trials separately are needed in order to show the differentiation in the first block in the Placebo group. We have added these plots (see figure below) to the supplementary material (figure S4). These plots show higher responses to observational US as compared to no observational US trials in the first block across groups (see figure below, a) for thalamic responses, b) for PAG responses and c) for amygdala responses).

Additionally, we ran a new first level analysis in order to examine the observational US responses in the first block in the Placebo group only. This analysis should differentiate between the observational US and no observational US trials (obs US > no obs US) in the first Block as a “normal function of observational fear conditioning” in the Placebo group. Indeed, this contrast revealed differences in the PAG and midline Thalamus in close proximity (i.e. 4mm spheres) to the results that are reported as group differences in our manuscript: PAG: x,y,z (NMI): -8,-28,-8; t=2.94; p(FWE)=0.026; midline thalamus: x,y,z (NMI): -6,-26,4; t=2.38 p(FWE)=0.085, see figure below panel a & b]. Moreover, these responses (obs US > no obs US) were similar across groups [effects across group PAG: x,y,z (NMI): -8,-28,-8; p(FWE)=0.030; midline thalamus: x,y,z (NMI): -6,-26,4; p(FWE)=0.002; see figure below panel c & d]. Hence, the difference between groups in the PAG and the midline thalamus towards the observational US (obs US > no obs US) reflects a normal brain response during observational fear conditioning that is altered through opioid receptor blockade.

Based on our analyses, we are confident that the effects observed in the amygdala, PAG, and the midline thalamus, represent responses during social threat learning under normal conditions, and that these are altered through the blockade of opioid receptors.

4) Related to the above (point 4), it is also unclear whether there is a significant training induced reduction in the observational US response in PAG and thalamus in the placebo group (Fig. 3b-c). Again, this is important to establish that they are seeing some kind of observational US prediction error response in these regions as they suggest.

Response:

We followed the reviewer's suggestion to focus on the learning induced changes of the PAG and thalamic responses in our analysis. Already in our first version of the manuscript, we reported a significant interaction between group and block, revealing that responses in the Placebo group decreased during learning, as compared to the sustained responses in the Naltrexone group. Importantly, across groups, there was a significant observational US response (differentiation between observational US > no observational US trials) in the PAG and trend-wise in the thalamus in the first block (PAG: paired t-test $t(42):2.25, p=0.030$; thalamus: paired t-test $t(42):1.70, p=0.096$), and this observational US response was not different between groups (PAG: independent t-test $t(42)<1, p>0.7$; thalamus: independent t-test $t(42)<1, p>0.9$). In the Placebo group, observational US responses (differentiation between observational US > no

observational US trials) decreased trend-wise from block 1 to block 2 in the PAG (paired t-test $t(20):1.95$, $p=0.065$) and significantly from block 1 to block 3 (PAG: paired t-test $t(20):2.12$, $p=0.047$). These results mirror our analyses above, showing that responses in the PAG represent a “normal” function in the beginning of the experiment that is altered in its time-course through the blockade of opioid receptors.

More specifically, we report in our first version of the manuscript the results of the logistic regression model, in which the PAG responses predict the SCR responses to the CS+ during observational fear conditioning. This analysis reveals a main effect for the PAG responses as a predictor for SCR responses in both groups, which is also trend-wise significant when the model is estimated for the Placebo group, only ($F: 2.72$; $p=0.09$). This model shows that PAG responses decrease, as SCR to the CS+ increases (see plot below).

To further address the reviewer’s comment about an observational US prediction error, we conducted additional, exploratory, analyses. First, we calculated a first level that models simplified prediction error responses, defined as the deviation between the outcome and the expected outcome. This Prediction error was modelled as absolute difference between the observed outcome of CS+ trials (observational US = 1/ no obs US = 0) and the sum of previous outcomes divided through the trial-numbers (i.e. average of outcomes of previous trials). The prediction error term was added as a parametric modulator of CS+ outcomes (controlling for the general outcome, i.e. obs US and no obs US).

A one sample t-test of activity in the Placebo group revealed significant activity in the PAG (see figure below) reflecting the time-course of the prediction error (overlap with PAG activity in the manuscript $x;y;z: -8;-30;-10;t=3.77$; $p(SVC): 0.030$; 4mm sphere).

Other regions, such as the medial thalamus, medial PFC and the amygdala also followed this time-course, which is in accordance with findings of neural correlates of Prediction errors in humans and animals. The same region in the PAG was stronger correlated with this prediction error time-course in the Placebo group as compared to the Naltrexone group, which did not follow a Prediction error related time-course (x;y;z: -8;-30;-10;t=3.21; p(SVC):0.020, see figure below).

We have added this prediction error analysis to the revised supplement (see “**Additional temporal modelling of PAG responses towards the observational US**”).

Hence, our data suggest that the decrease in the PAG in the Placebo group is indeed related to learning progress in the Placebo group and that the blockade of opioid receptors changes this PAG time-course. Future studies that are designed to accurately model behavioural responses (e.g. determine Expected value and individual learning rates) might reveal the neuro-computational underpinnings of social threat learning, and the influence of the opioid system on the learning processes in more detail.

5) In the cross-regional interaction analyses in Fig. 3d the authors say that “PAG responses displayed an increased functional connectivity (PPI) with theSTS”. From my reading of the paper and Methods it is not clear what responses they are analyzing here. Observational US? CS+?

Response:

The PPI examined functional connectivity of the PAG during the observational US (again obs US > no obs US contrast). The reviewer’s comment lead us to realize that we need to state this more clearly, and we have therefore changed the description of the PPI methods. We now state (page 5): “In order to test if the temporal dynamic of PAG responses towards the observational US were functionally connected with other brain regions in the Naltrexone group, we compared condition specific connectivity (psycho-physiological interaction, PPI, see supplementary methods) between groups”

Moreover, in addition to the information about the PPI already present in the supplementary methods, we have added the following to the main text (Page 8):

“Psycho-physiological interaction (PPI, as implemented in SPM8, see supplementary methods for details) was used to examine functional connectivity differences of PAG responses towards the observational US (observational US > no observational US) between groups. Extracted eigenvariates of the PAG peak voxel were used as the seed region, deconvolved and controlled for the PAG time-course and the onset regressor.”

6) In the decoding analysis presented in Fig. 4, they use their supervised machine learning approach to decode treatment group from the CS+ responses during the immediate expression test. As they saw no behavioral differences (but see point 1) at this timepoint it is not clear why they are looking here rather than at the long term test timepoint. This should at least be discussed if not extended to the later timepoint.

Response:

We agree with the reviewer that fMRI data during the long-term test would have been beneficial, however this test was performed in the behavioural lab, only. We have added a sentence in the main text stating explicitly that no behavioural effect was observed in the immediate test stage, which was used for decoding (Page 6): “While we found no difference in threat expression between groups in the immediate test in the SCRs, this result might point towards a difference in brain activation patterns between groups during threat expression.”

Additionally, we explored if the individual functional weights in the decoding analysis are correlated with the threat expression 72 hours later, but found no significant association in the Placebo or the Naltrexone group (both $p > 0.7$).

7) Also related to the decoding analysis, the authors should examine individual brain regions for classification analysis (in conjunction with the whole network analysis that they use now) to determine which brain regions are most important for the decoding.

Response:

We thank the reviewer for this suggestion. This analysis revealed that bilateral anterior temporal regions contributed the highest weights for classification. We include this information in the revised manuscript on page 6: “Computation of functional weights revealed the highest values for anterior temporal regions in proximity to the amygdala, including the bilateral anterior temporal gyrus and the left temporal pole. Additionally, the right caudate and right thalamus contributed high weights, as well (see **figure 4** and **table S9**).”

8) Related more generally to the effect of naltrexone on brain responding, it is possible that the drug is changing the brain in some way that it appears more active than in placebo conditions (ex. Changes in resting activity or general stimulus evoked activity). This could explain the effects US processing, cross-region coupling and drug treatment decoding they observe. The authors should try to address this in some way.

Response:

We agree with the reviewer that pharmacological manipulations runs the risk of confounding the results due to its physiological effects unrelated to the paradigm. To avoid such confounds, we have used a differential paradigm (including CS+ vs. CS- and obs US vs. no obs comparisons) in the fMRI and behavioural analyses. Our design thus addresses these concerns by providing within-subject, and within-sessions, control events. Yet, to further bolster our conclusions and address the reviewer's concerns, we analyzed responses during the ITI, which should be unrelated to (but not completely independent from) the learning task. The comparison between groups revealed no differences between groups in our ROIs (see the figures below, representing the Contrast of Naltrexone > Placebo in the amygdala and the PAG/mid thalamus). Hence, although activity during the ITI might reflect some aspects of anxiety, there was no significant difference between groups. This renders the hypothesis unlikely that the results obtained in our study represent a general effect of Naltrexone on the physiological responses in the brain.

Minor Points

1) They reference Table S1 on pg 2 in support of the idea that the amygdala is activated by observational USs, but from my reading it looks like Table S1 only deals with PAG responses.

Response:

We apologize for accidentally leaving these responses out. We have now corrected this in the new version of the manuscript.

2) The figure legend for Fig. 3 discusses a panel 'e' which is not in the figure.

Response:

The reviewer is right that panel e was blended with panel d. This has now been changed.

Reviewer #3 (Remarks to the Author):

In this manuscript, the authors examine the hypothesis that opioidergic neural circuits shape prediction error during observational threat conditioning in humans. The results show that the opioid antagonist naltrexone, given prior to observational fear learning, enhances amygdala activity to the observed US, and also produces a correlation between amygdala activity to the US during learning and the degree of long-term memory measured 3 days later. The authors also report clusters of activity evoked by the observed US in the midline thalamus and PAG that are greater for the naltrexone group compared to the placebo controls.

The findings of this study are largely confirmatory. That is, they show that neural fear

circuits identified in animal studies are also important in humans. The major contribution here is showing that opioids are important for observational fear memory in humans. However, the role of endogenous opioids seems to eliminate, rather than “limit” (as claimed by the authors), observational fear because most subjects in the placebo control group did not have long-term observational fear memory. Overall, I find the question to be an interesting one, but I’m not sure that the findings reach the novelty level required for Nature Communications.

Response:

We thank the reviewer’s for his/her comments. We are delighted that the reviewer shares our conviction that the “..major contribution here is showing that opioids are important for observational fear memory in humans”. Indeed, we believe that the demonstration that the opioidergic circuit, which has been extensively studied in animals during direct aversive learning, is involved in humans during observational fear learning, is important. Please see our replies to the reviewers` comments below. In order to facilitate the review process, we have marked changes in the main text in yellow.

1) The authors note that the groups, on average, show evidence of learning (Figure S1). Yet, it seems clear from Figure 1b that a substantial number of participants in both the placebo and naltrexone groups must not have learned the task. In this graph, a negative difference between the CS+ and CS- indicates that a subject had a greater SCR for the CS- than the CS+. It is not clear to me why these subjects who did not learn the task would be included in any analysis. Also, it appears that, on average, participants in the placebo group (Figure 1a) didn’t show long-term memory, because the average SCR difference score was negative. How can the authors claim that opioid receptor blockade “enhanced observational fear learning” if there was no observational fear learning in the controls (placebo group)?

Response:

The reviewer is right, that some individuals in the Placebo group showed a negative difference between the CS+ and the CS- at the long-term test. However, the Placebo group, as well as the Naltrexone group, showed descriptively higher responses to the CS+ as compared to the CS- at the first trial of the long-term test (see figure below). This might suggest that both groups initially retrieved the CS-US association, yet this association was more persistent in the Naltrexone group.

Moreover, it is not inherently problematic that the Placebo group showed lower expression of fear conditioned responses at the long-term test stage. As the reviewer noted, the immediate test stage (figure S1) revealed successful expression of conditioned responses in both groups. Importantly, during this immediate test stage, the conditioned responses are extinguished to some degree (due to non-reinforcement). Hence, the long-term test stage examines the persistence of the conditioned responses, and is not a test of whether the groups learned in the first place. We think that we can make this point more clearly in the description of our paradigm, and have therefore added the following to the revised manuscript: (page 2)

“During both the immediate, and long-term, expression test, CSs were presented directly to the participants in absence of the demonstrator and never followed by a US. Therefore, conditioned responses might have extinguish during the test stages. While the immediate test stage allows us to test the expression of learning immediately after acquisition, the long-term test stage examines the persistence/return of the acquired threat associations that are learned via observation.”

2) It is not clear from the text whether the analyses restricted to the “observed US” period only used data from the 12 trials in which the observed US was actually presented. Please clarify.

Response:

The analyses of responses towards the observational US examined responses to the 12 CS+ trial outcomes in which an observational US was presented. These responses were contrasted with CS+ trial outcomes when no observational US was presented (i.e. observational US > no observational US). This comparison allowed us to contrast responses to the outcomes that were not biased by the presence of the CS that was shown before these events. We apologize if this was not clear in our initial version of the manuscript. Please, see our response to the next comment for the changes in the

revised version.

3) Figure 3b It isn't clear what the "no obs US" trials are. Are these the CS+ trials without any US presented? Are they the CS- trials? This type of trial-by-trial analysis should be shown for the amygdala as well.

Response:

We apologize for being unclear on this point. Indeed, the reviewer is right that the "no obs US" responses are CS+ trials that were not followed by an observational US (see comment above). We have made this more clearly in the main text of the revised manuscript (Page 2 and 3):

"For that purpose, we contrasted responses across both groups towards the observational US (occurring at the end of 50% of the CS+ trials, termed *obs US*) with responses during the same time-point to CS+ trials not followed by the US (termed *no obs US*), which controls for the influence of the preceding CS on observational US responses".

Moreover, we added the following information to the figure captions of figure 2 and 3. "Obs US" refers to responses to the observational US and "no obs US" to responses to CS+ outcomes that are not followed by the US."

Additionally, as suggested by the reviewer, we have now added to the main text the time-course analyses of amygdala activity (Page 5):

"The ANOVA of the extracted responses in the midline thalamus and the left amygdala did not reach significance [Block by group interaction: mid thalamus $F(2,82)=2.3;p=0.10$; left amygdala $F(2,82)=2.7;p=0.08$), however block-wise comparisons between groups, revealed higher responses in the Naltrexone group towards the observational US in block 2 in both regions (mid thalamus t-test two-tailed $p=0.002$; left amygdala t-test two-tailed $p=0.004$) and trend-wise in block 3 in the thalamus only."

In addition, we have provided a figure (see figure panel c) below) displaying the specific responses to the obs US and no obs US trials in the supplement (figure S4).

4) The Figure 4 legends references error bars, but no error bars are depicted in the graphs.

Response:

The reviewer is completely right. We have now changed this mistake in the revised manuscript.

REVIEWERS' COMMENTS:

Reviewer #1 (Remarks to the Author):

The authors have addressed my comments on the initial version of this manuscript via significant text revisions and inclusion of new data. I view it as improved and there remains much to enjoy about this manuscript. However, there are still some aspects that I still struggle with.

Most importantly, the localisation of the PAG still concerns me. The authors have given a clearer statement of how they defined the PAG, and this rests on a paper by Fairhurst et al. My difficulty is that when I examine the figures, I do not really see a BOLD signal in PAG. Figure 3A shows a sagittal section with overlaid BOLD responses, and the bottom inset in that figure purports to localise the BOLD response to PAG. I simply can not see how the highlighted region is PAG: the region highlighted is well lateral to PAG, possibly it is the deeper layers of the colliculus, it is difficult to tell. This is not an uncommon problem when imaging the human brainstem, and perhaps the simplest way to address this would be to acknowledge this in the Discussion.

[**Editorial Note:** The editors judged this concern to be sufficiently important and asked all three reviewers to comment and for authors to respond.]

Reviewer #2 (Remarks to the Author):

I understand the Reviewer 1's concern. The authors could potentially address this concern by putting together a detailed figure showing the BOLD activation (or subtraction) for each subject along with a circle showing denoting the coordinates listed in the Methods (x: +/- 6; y: -34; z: -6). This could be included in the Supplementary Information.

Reviewer #3 (Remarks to the Author):

I agree that the figure panel is more consistent with the colliculus than PAG. After re-reading the paper, I was trying to figure out the number of voxels that corresponded to the PAG, and I realized that the authors did not report their acquisition parameters in sufficient detail--they report only the voxel size after re-slicing. Please report the original scan parameters (including voxel size) in the Methods.

We were extremely delighted to learn about the reviewers' favourable evaluation of our revised manuscript. We take the remaining concern about the localization of the PAG very seriously. Fortunately, we feel that we can fully address this concern, which we do in two principal ways: first, by providing more evidence for that the reported activation lies within the PAG, and secondly, by carefully acknowledging the possibility of co-activity in neighbouring structures (colliculus). Here below, we insert our responses to the reviewers' comments.

Reviewer 1's comments:

The authors have addressed my comments on the initial version of this manuscript via significant text revisions and inclusion of new data. I view it as improved and there remains much to enjoy about this manuscript. However, there are still some aspects that i still struggle with.

Most importantly, the localisation of the PAG still concerns me. The authors have given a clearer statement of how they defined the PAG, and this rests on a paper by Fairhurst et al. My difficulty is that when I examine the figures, I do not really see a BOLD signal in PAG. Figure 3A shows a sagittal section with over layed BOLD responses, and the bottom inset in that figure purports to localise the BOLD response to PAG. I simply can not see how the highlighted region is PAG: the region highlighted is well lateral to PAG, possibly it is the deeper layers of the colliculus, it is difficult to tell. This is not an uncommon problem when imaging the human brainstem, and perhaps the simplest way to address this would be to acknowledge this in the Discussion.

Authors Response:

We agree with the need to more clearly acknowledge in the discussion (and the results section) the possibility that neighbouring structures (here the colliculus) might show co-activations. The colliculus is indeed an input and output region of the lateral PAG, and is therefore likely to display a co-activation. To acknowledge this, we have added (highlighted in yellow) to the results section (page 3):

"We contrasted hemodynamic activity towards the observational US between groups (Naltrexone > Placebo), which revealed higher responses to the observational US in the Naltrexone as compared to the Placebo group within the PAG ROI (left x,y,z (MNI)=-8;-32;- 8; t=3.15;p(SVC)=0.016, see figure 3a,b). This activation was located

on the left side ventrally to the central aqueduct, most likely located in the PAG (see supplementary figure S7) and extending into the colliculus."

and in the discussion (page 7):

"Additionally, our results revealed co-activity in a neighbouring structures of the PAG, including the colliculus, which has been described as both an output and input region to the PAG (CARRIVE and MORGAN, 2004). Future research is warranted to employ high resolution of the brainstem function in observational learning in order to describe the contribution of the PAG and neighbouring regions in greater details."

Additionally, we agree with the reviewer that the sagittal view in figure 3a is not optimal to show the location of the PAG. This activation map was chosen to illustrate all activity within the midbrain and thalamic structures. In order to enhance clarity, we have included a new supplementary figure that provides more detailed information about the location of the effect in the PAG. Reviewer 1 highlights that we base the location of the PAG on one study (Fairhurst et al.). We agree that a more careful evaluation of the PAG location is warranted to better support our claims. In order to provide stronger evidence of that the activations that we report indeed are located within the PAG, we used new coordinates from a metaanalysis (Linnman et al. 2012, Neuroimage) including 225 published reports on the location of the PAG. The meta-analysis found the left PAG to be located on average within the following box (\pm SD) at $x:-4(\pm 3), y:-29(\pm 5), z:-12(\pm 7)$. The result reported in our manuscript, representing the difference in the US responses between Naltrexone and Placebo, lies within this box ($p(\text{SVC})=0.026; t=3.55; x:-7; y:-32; z:-8$). This puts our results into the context of published research on PAG responses in humans, verifying that our definition of PAG responses aligns with previous research.

In order to illustrate these results in a better fashion, we have added the following figure S7 to the supplementary materials to display in insert a) how the activations in our results are mapping on the average location (\pm SD) of the left PAG as defined in a Meta-analysis by Linnman et al. 2012.

Figure S7: Higher Responses to the observational US (obs US > no obs US) in the Naltrexone group as compared to Placebo. a) The average group difference is located within an average location (\pm SD) of the left PAG as defined in a Metaanalysis by Linnman et al. 2012 (indicated by the red line). b-d) Location of the maxima of individual effect sizes (each square represents a participant) within this

average PAG location revealed majorly activity close to the central aqueduct, and some maximal effects in neighbouring regions.

In light of these new pieces of information, we are confident that the differences between groups represent activity within a structure located within the PAG. Moreover, we hope that our added discussion about possible co-activations of neighbouring structures caution against overinterpreting our neuroimaging results of a small structure, such as the PAG.

Reviewer 2's comments on this concern:

I understand the Reviewer 1's concern. The authors could potentially address this concern by putting together a detailed figure showing the BOLD activation (or subtraction) for each subject along with a circle showing denoting the coordinates listed in the Methods (x: +/- 6; y: -34; z: -6). This could be included in the Supplementary Information.

Authors Response:

We followed the suggestion of Reviewer 2 and inserted a display of the individual maximal effect within the PAG (as defined by a meta-analysis by Linnman et al. 2012, see above) to the supplement.

Figure S7: Higher Responses to the observational US (obs US > no obs US) in the Naltrexone group as compared to Placebo. a) The average group difference is located within an average location (+/- SD) of the left PAG as defined in a Metaanalysis by Linnman et al. 2012 (indicated by the red line). b-d) Location of the maxima of individual effect sizes (each square represents a participant) within this

average PAG location revealed majorly activity close to the central aqueduct, and some maximal effects in neighbouring regions.

We thank the reviewer for this important suggestion, and believe that our reply provide additional evidence of that the effect lies within the PAG, as well as involves co-activity in neighbouring regions (which we carefully discuss in the results and discussion, see reply above).

Reviewer 3's comments on this concern:

I agree that the figure panel is more consistent with the colliculus than PAG. After re-reading the paper, I was trying to figure out the number of voxels that corresponded to the PAG, and I realized that the authors did not report their acquisition parameters in sufficient detail--they report only the voxel size after re-slicing. Please report the original scan parameters (including voxel size) in the Methods.

Authors Responses:

We agree with the Reviewer (as in our reply to Reviewer 1) that the figure panel is not optimal in the sagittal view. This view was intended to display the midbrain activity and thalamic responses equally well. Therefore we have now included a supplementary figure (S7, see responses above) illustrating the location of PAG responses in relation to a mask derived from a meta-analysis of 225 PAG locations (see our reply to Reviewer 1 above), as well as the location of the individual maximum peak within this mask.

Regarding the second point, we are sorry for mistakenly not reporting the initially acquired voxel size. We have added the following text marked in yellow to the revised methods:

"Functional magnetic resonance imaging (fMRI) data was acquired using a 3 Tesla MR scanner (General Electrics 750) with an 8-channel head coil. Each functional image volume comprised 47 continuous axial slices (3 mm thick, 0.7 mm gap) that were acquired using a T2*-sensitive gradient echo-planar imaging (EPI) sequence [repetition time (TR): 2870 ms; echo time (TE): 30 ms; flip angle: 90°; 2.3 x 2.3 mm in-plane resolution]. The first 5 volumes of each time series were discarded to account for T1 equilibrium effects. Pre-processing involved distortion correction of susceptibility-induced gradients of BOLD images through field-maps, realignment, unwarping, co-registration and normalization to a sample-specific template using DARTEL and spatial smoothing (6 mm FWHM isotropic Gaussian kernel) within the "Statistical parametric mapping" (SPM8, www.fil.ion.ucl.ac.uk/spm) software package."

Reviewer #1 (Remarks to the Author):

The authors have addressed my comments on the initial version of this manuscript via significant text revisions and inclusion of new data. I view it as improved and there remains much to enjoy about this manuscript.

These additional PAG data are very helpful indeed. These are difficult experiments and imaging human brainstem is simply a nightmare. I think the new analyses and the new figures, plus the new text, are all that I could expect to adequately address this issue using existing technologies.

Reviewer #2 (Remarks to the Author):

The authors have adequately addressed my concerns. I endorse publication.

The authors might also think about adding a citation to a recently published paper dealing with the role of the amygdala and PAG in setting aversive prediction error coding during first order fear conditioning which has direct relevance to this study:

Ozawa, T., Ycu, E.A., Kumar, A., Yeh, L-F., Ahmed, T., Koivumaa, J., and Johansen, J.P. A feedback neural circuit for calibrating aversive memory strength. *Nature Neuroscience* 2016, 20(1):90-97

Reviewer #3 (Remarks to the Author):

I am largely satisfied with the revisions made in response to my previous comments, and have only a few points for clarification:

- 1) What does the asterisk in Figure 2b represent?
- 2) In Figure 3c, the x-axis label may be in a different language ("a 4 trials")?
- 3) It isn't clear to me whether subjects thought they might ever receive a shock. I understand that they were hooked up to the shock apparatus, but what were the instructions they received? Were they instructed that they might receive shock, or were they instructed that they would not receive shock?

Reviewer #2 (Remarks to the Author):

The authors have adequately addressed my concerns. I endorse publication. The authors might also think about adding a citation to a recently published paper dealing with the role of the amygdala and PAG in setting aversive prediction error coding during first order fear conditioning which has direct relevance to this study:

Ozawa, T., Ycu, E.A., Kumar, A., Yeh, L-F., Ahmed, T., Koivumaa, J., and Johansen, J.P. A feedback neural circuit for calibrating aversive memory strength. *Nature Neuroscience* 2016, 20(1):90-97

Response: We thank the reviewer for this valuable suggestion. The publication is now cited in the discussion section (page 7): "Moreover, our finding that the temporal dynamics of PAG scales aversive learning from observation of others is consistent with a recent finding in animals, showing that prediction error coding in the PAG (and the amygdala), sets aversive memory strengths during learning from direct aversive experiences³²".

Reviewer #3 (Remarks to the Author):

I am largely satisfied with the revisions made in response to my previous comments, and have only a few points for clarification:

1) What does the asterisk in Figure 2b represent?

Response: The asterisks indicate statistically significant differences between groups. In particular, in figure 2b this asterisk represents the difference in correlation coefficients between groups. We have added this to the caption of figure 2: "Asterisks indicate significant differences between groups."

2) In Figure 3c, the x-axis label may be in a different language ("a 4 trials")?

Response: We wanted to indicate that one block represents 4 trials. We have made this explicitly clear in the re-worked figure 3.

3) It isn't clear to me whether subjects thought they might ever receive a shock. I understand that they were hooked up to the shock apparatus, but what were the instructions they received? Were they instructed that they might receive shock, or were they instructed that they would not receive shock?

Response: The participants were instructed that electrical stimulation would be possible during the experiment. We have now explicitly stated this in the methods: "Before starting the experimental task, participants were attached to SCR and shock electrodes with the instruction that they might receive a shock at any time during the whole time-course of the experiment."